# NEURAL FUNCTIONS FOR LEARNING PERIODIC SIGNAL

**Woojin Cho**[1*]    **Minju Jo**[2*]    **Kookjin Lee** [3]    **Noseong Park** [4]
[1]TelePIX,    [2]LG CNS,    [3]Arizona State University,    [4]KAIST
`{woojin.py, minnju42, kjlee8344}@gmail.com`,
`noseong@kaist.ac.kr`

## ABSTRACT

As function approximators, deep neural networks have served as an effective tool to represent various signal types. Recent approaches utilize multi-layer perceptrons (MLPs) to learn a nonlinear mapping from a coordinate to its corresponding signal, facilitating the learning of continuous neural representations from discrete data points. Despite notable successes in learning diverse signal types, coordinate-based MLPs often face issues of overfitting and limited generalizability beyond the training region, resulting in subpar extrapolation performance. This study addresses scenarios where the underlying true signals exhibit periodic properties, either spatially or temporally. We propose a novel network architecture, which extracts periodic patterns from measurements and leverages this information to represent the signal, thereby enhancing generalization and improving extrapolation performance. We demonstrate the efficacy of the proposed method through comprehensive experiments, including the learning of the periodic solutions for differential equations, and time series imputation (interpolation) and forecasting (extrapolation) on real-world datasets.

## 1 INTRODUCTION

Coordinate-based multi-layer perceptrons (MLPs) or implicit neural representations (INRs) learn a continuous representation of a signal, which are collected in discrete measurements, not necessarily sampled in a uniform mesh grid. Being independent on a regularly-sampled data and learning the continuous signal naturally address difficulties in model training on irregularly sampled data or data that are not on a Cartesian system. These unique features lead to many successes in applications that have historically posed difficulties for conventional approaches, including learning solutions of complex nonlinear differential equations (DEs) (Sitzmann et al., 2020; Raissi et al., 2019), 3D scene representations (Mildenhall et al., 2021), etc.

Recognizing the potential of INRs, extensive efforts have been dedicated to improving various aspects of INRs. Notably, in Sitzmann et al. (2020), a new activation function has been proposed to facilitate the learning of high-frequency signals. In Dupont et al. (2022); Lee et al. (2021); Cho et al. (2024), meta-learning-based training algorithms have been studied for efficiency. In De Luigi et al. (2023); Zhou et al. (2023), research has explored methods for navigating a learned latent space and extracting hidden representations to enhance downstream task performance.

Despite these recent advancements, even in its state-of-the-art (SoTA) formulation, INRs surprisingly fall short in effectively representing signals that exhibit a sequential nature (such as time series). This deficiency becomes particularly evident in their inability to accurately predict signals at out-of-distribution (OoD) inputs, leading to catastrophic failures in extrapolation or forecasting scenarios. Figure 1 visually illustrates an example of learning a simple periodic function $\sin(50x)$ with recent INR architectures including SIREN (Sitzmann et al., 2020), FFN (Tancik et al., 2020), and WIRE (Saragadam et al., 2023); the example essentially demonstrates that those existing architectures struggle in extrapolation.

---

[*]Equal Contribution

In this study, we focus on scenarios where the ground truth signal is exhibiting periodicity, which is indeed common in real-world scenarios; for example, in natural science, periodic behaviors are observed in many applications from a simple example, such as the oscillation of a pendulum, which is a classic example of periodic motion governed by principles of physics, to a complex example such as Earth's periodic and cyclical variations of weather and temperature. Consequently, capturing periodicity from a given range of data (e.g., historical data) is critical in accurate prediction in extrapolation (e.g., future evolution).

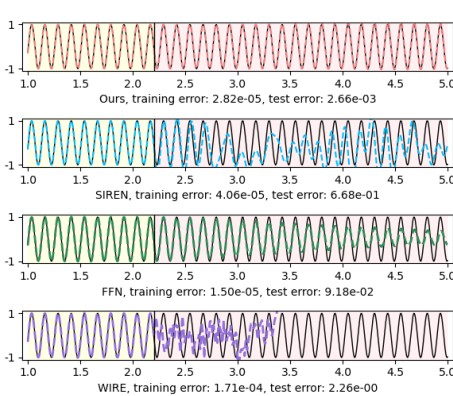

Figure 1: Regression of a target function $\sin(50x)$ and extrapolation tests. The training (yellow) and the test (red) regions are separated by the vertical bar ($x = 2.2$).

To this end, we propose a novel architectural design of INRs. The design involves two key aspects: i) learning Fourier feature mappings for transforming a spatial and/or temporal coordinate to facilitate capturing of periodicity and ii) decomposing the learning of periodic and scale components of an observed signal. This separation introduces inductive bias of the existence of the periodicity and the trend (dictated by the scale component), enabling precise prediction in extrapolation settings. Moreover, the proposed model inherits all desirable traits of conventional INRs: the capability to operate on irregularly-sampled data, per-data-instance-basis (eliminating the need for a large dataset for training), and a lightweight structure (no need for sliding windows or heavy numerical computations during inference).

In summary, our contributions include:

1. We design a novel INR architecture employing two separate internal networks for learning periodic and scale components of an observed signal.
2. We propose an algorithm, which requires only a single trained model to perform both interpolation (imputation) and extrapolation (forecasting).
3. We demonstrate that our design outperform SoTA baselines in interpolation and extrapolation tasks in several well-known real-world benchmark problems.

## 2 DESIDERATA FOR INR-BASED MODELING FOR PERIODIC FUNCTIONS

Now, we identify some desired characteristics required for INRs in handling periodic sequential data, and based on the identified characteristics we design an efficient and effective network.

### 2.1 DECOMPOSITION OF SEQUENTIAL SIGNAL INTO FACTORS

Decomposing a time series signal into interpretable factors, e.g., seasonality and trend, is a common practice for improving the performance of models in the field of time series (Cleveland et al., 1990) (cf. Section 5). Separately predicting them and later combining them into a signal is effective in improving the time series model accuracy (Hamilton, 2020). We propose a new INR design for separately modeling the periodic factor and the scale factor and later combining them to reconstruct the signal. This factor-by-factor processing approach alleviates the modeling burden of INRs for periodic sequential data. Also, we do not explicitly supervise the learning of the two individual factors; instead we train the proposed model with the original undecomposed signal to reduce the training overhead and implicitly learn the two factors. Our theoretical analysis (cf. Appendix D) shows that learning time series through this design allows effectively extracting periodic factors.

### 2.2 LEARNING FOURIER FEATURES

Fourier features of signals introduce a new angle, interpreting the signals in the frequency domain, which is known to be effective in time series modeling. Thus, using Fourier features in the network design can be beneficial for sequential data modeling. Recent INRs, such as SIREN (Sitzmann et al.,

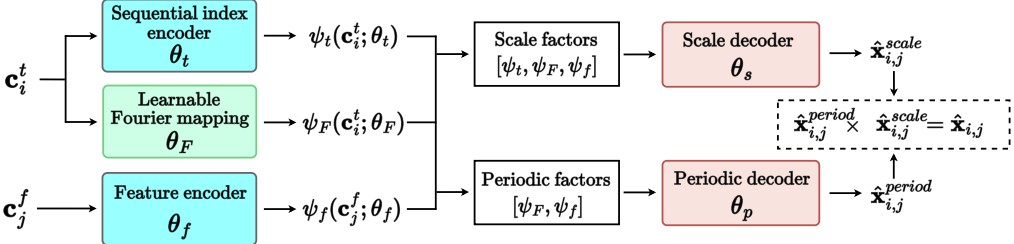

Figure 2: **NeRT architecture.** The input coordinate $(\mathbf{c}_i^t, \mathbf{c}_j^f)$ is converted to periodic/scale factors through $\psi_t$, $\psi_F$, and $\psi_f$, followed by two decoders $\phi_s$ and $\phi_p$ to effectively infer the signal intensity $\mathbf{x}_{i,j}$ at the input coordinate.

2020) and FFN (Tancik et al., 2020), also aggressively use Fourier features. However, those existing Fourier feature extraction methods are optimized mainly toward images (e.g., a hand-tuned frequency in the input layer of SIREN) and they often do not show satisfactory performance for sequential data modeling (e.g., poor extrapolation shown in Figure 1). Taking a cue from Li et al. (2021), we overcome the limitation by learning a Fourier feature mapping layer (like FFNs but we learn a Fourier mapping instead of the FFNs' hand-craft mapping) and utilizing the sinusoidal activation (like SIRENs) — the details of our model design for NeRT will be described shortly in the next section and our theoretical analyses justify the appropriateness of our design to sequential data processing.

## 3 MODEL ARCHITECTURE

We describe our proposed INR-based framework (cf. Figure 2), **NeRT**, for interpolation and extrapolation on functions with periodicity. As we intend to learn a periodic sequential signal in a factored form, a periodic factor and a scale factor, we design an encoder-decoder-type neural network architecture, where i) the encoders generate embeddings for input coordinates, and ii) the decoders take the embeddings as input and produce a periodic factor and a scale factor, individually. The two factors are then multiplied to produce the signal at the input coordinate.

### 3.1 ENCODER

The encoder of NeRT reads an input coordinate of multi-dimensional sequences, $\mathbf{c}_i = (\mathbf{c}_i^t, (\{\mathbf{c}_j^f\}_{j=1}^M)), i = 1, \ldots, N$, where $N$ and $M$ denote the length and the dimensionality of the sequences. The first coordinate $\mathbf{c}_i^t$ refers to the sequential index and the second coordinate $\mathbf{c}_j^f$ refers to an one-hot vector with one in only the $j$th element, which is used to index the $j$th feature. Depending on the application, the first coordinate $\mathbf{c}_i^t$ is typically a temporal index $t$ in case of learning time series data or a spatial index $x$ in case of learning a periodic signal defined over a spatial domain. We further describe other encoding schemes for processing raw coordinate input in Appendix C.

**Embedding of the Input Coordinate**  For an input coordinate $(\mathbf{c}_i^t, \mathbf{c}_j^f)$, we use the following FC layers with ReLU activation functions:

$$\psi_t(\mathbf{c}_i^t; \theta_t) := FC_{L_t} \circ \rho_r \circ FC_{L_t-1} \circ \cdots \circ \rho_r \circ FC_1(\mathbf{c}_i^t),$$
$$\psi_f(\mathbf{c}_j^f; \theta_f) := FC_{L_f} \circ \rho_r \circ FC_{L_f-1} \circ \cdots \circ \rho_r \circ FC_1(\mathbf{c}_j^f), \tag{1}$$

where $L_t$ and $L_f$ mean the number of layers in each encoder respectively, and $\rho_r$ is ReLU. Since $\mathbf{c}_j^f$ is an one-hot vector, one can consider that $\psi_f(\mathbf{c}_j^f; \theta_f)$ generates an embedding for the one-hot vector.

**Learnable Fourier Feature Mapping**  The previous two embeddings, $\psi_t$ and $\psi_f$, do not generate Fourier features and therefore, we use the following method to generate them. In particular, our method is to learn an optimal Fourier mapping layer (instead of hand-crafted ones):

$$\psi_F(\mathbf{c}_i^t; \theta_F) := \{\mathbf{A}_m \odot \sin(\boldsymbol{\omega}_m \mathbf{c}_{i,m}^t + \delta_m) + \mathbf{B}_m\}_{m=1}^{D_{\mathbf{c}^t}}, \tag{2}$$

where $\mathbf{c}^t_{i,m}$ is $m$-th value of $\mathbf{c}^t_i$ and $\odot$ denotes element-wise multiplication. $\boldsymbol{\omega}_m \in \mathbb{R}^{1 \times D_{\psi_F}}$ is the $m$-th frequency of the Fourier feature $\psi_F$, which is sampled from a uniform distribution $[a, b]$. A learnable vector $\mathbf{A}_m \in \mathbb{R}^{1 \times D_{\psi_F}}$ indicates the amplitude, initialized to 1. Vectors $\mathbf{B}_m \in \mathbb{R}^{1 \times D_{\psi_F}}, \delta_m \in \mathbb{R}^{1 \times D_{\psi_F}}$ denote the phase shift and the bias respectively, and they are initialized to 0. Optionally, we can employ an additional FC layer after Equation 2 to compress the embedding. All together, we use three types of embeddings, $\psi_t$, $\psi_f$, and $\psi_F$ given the coordinate.

*Remark* 3.1. According to the NTK theory (Jacot et al., 2018), our kernel satisfies the stationary and the shift-invariant properties — thus, one can consider that the learnable Fourier feature mapping of NeRT performs a 1D convolution-based processing of signal with a learnable kernel. In addition, our proposed mapping has an additional property that resorts to the extreme value theorem to perform prediction in OoD (extrapolation) — $\psi_F$ is i) continuous, and ii) confined to the min/max values defined by $\mathbf{A}$.

## 3.2 DECODER

The proposed decoder architecture separately generates the periodic and the scale factors, denoted $\phi_p$ and $\phi_s$, and multiply them to infer the signal intensity:

$$
\begin{aligned}
\phi_p(\psi_F \oplus \psi_f; \theta_p) &:= \rho_s \circ FC_{L_p} \circ \rho_s \circ FC_{L_p-1} \circ \cdots \circ \rho_s \circ FC_1(\psi_F \oplus \psi_f), \\
\phi_s(\psi_F \oplus \psi_f \oplus \psi_t; \theta_s) &:= FC_{L_s} \circ \rho_r \circ FC_{L_s-1} \circ \cdots \circ \rho_r \circ FC_1(\psi_F \oplus \psi_f \oplus \psi_t),
\end{aligned}
\tag{3}
$$

where $L_p$ and $L_s$ are the numbers of layers in the two decoders, $\rho_s$ is the sinusoidal function, and $\oplus$ denotes a concatenation of vectors. The output of $\phi_p$ and $\phi_s$ correspond to $\hat{\mathbf{x}}^{period}$ and $\hat{\mathbf{x}}^{scale}$ in Figure 2. We constrain $\hat{\mathbf{x}}^{period}_{i,j} \in [-1, 1]$ by using the Sine activation and $\hat{\mathbf{x}}^{scale}_{i,j} \in \mathbb{R}$ to denote the inferred periodic and scale factors, respectively. Note that the decoder to infer the periodic factor, i.e., $\phi_p$, mainly rely on our Fourier feature $\psi_F$; since $\psi_f$ is the embedding of the spatial coordinate, only $\psi_F$ contains the temporal information to infer. The scale decoder reads all available embeddings for the input coordinate $(\mathbf{c}^t_i, \mathbf{c}^f_j)$. Our final inference outcome, i.e., the signal intensity, at the input coordinate is $\hat{\mathbf{x}}_{i,j} = \hat{\mathbf{x}}^{period}_{i,j} \times \hat{\mathbf{x}}^{scale}_{i,j}$.

The advantages of our design are twofold: i) Our learnable Fourier feature mapping layer is specialized to extract periodic features according to our NTK-based analysis and, thus, it is sensible to model the periodic factor of the inferred signal intensity based on the Fourier feature and ii) the periodic and scale factors give us interpretable predictions for sequential signals both in interpolation and extrapolation.

## 3.3 TRAINING ALGORITHM

To train, we employ a regular gradient-based optimizer to minimize a mean-squared error (MSE) over data and predictions, $\frac{1}{MN} \sum_{i=1}^{N} \sum_{j=1}^{M} (\mathbf{x}_{i,j} - \hat{\mathbf{x}}_{i,j})^2$. Here, we consider $M$-variate sequences measured at $N$ collocation points (See Appendix F for a pseudo-code like algorithm). We emphasize again that the proposed model requires a single model training and use the same trained model to perform both interpolation and extrapolation while other non-INR baselines require training of different models for each task.

**Regularizing Scale Component**   Imposing an extra constraint on the scale component such as a penalty on the magnitude of the 3rd-order derivative of the scale w.r.t. the input can regularize the output of the scale component, preventing it from learning oscillatory patterns from the signal. Thanks to the inherent nature of INRs, any order of derivative can be computed via automatic differentiation. In the experiments section, we present results with such penalty.

**Modulation**   For training INRs for multiple sequence instances, we provide two different options: a *vanilla* mode or a *modulated* mode. For the modulation, we adopt an idea of *latent modulation* to our NeRT in Appendix K. To summarize, in the vanilla mode INRs are trained individually and in the modulated mode the shared part is trained via a meta-learning algorithm.

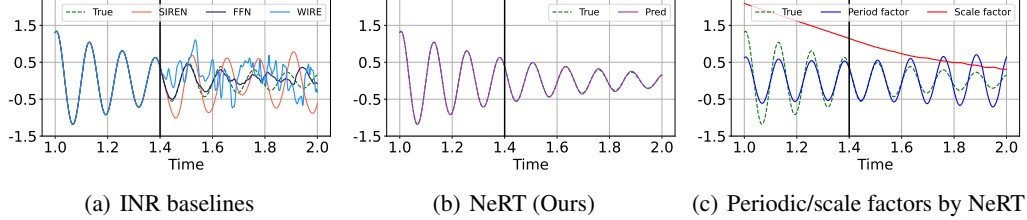

(a) INR baselines     (b) NeRT (Ours)     (c) Periodic/scale factors by NeRT

Figure 3: **Damped oscillator.** Extrapolation results (Figures 3 (a)-(b)) and extracted factors during training (Figure 3 (c)). The left side of the solid vertical line represents the training range, while the right side represents the testing range.

## 4 EXPERIMENTS

In this section, we evaluate the performance of NeRT on several benchmark problems. We have largely two sets of experiments: Synthetic data and real-world datasets. In the first set of experiments, we consider synthetic periodic data obtained from solutions of ODEs/PDEs, showcasing the model's applicability to different domains, i.e., spatial (PDE) or temporal (ODE) domains. In the next set of experiments, we demonstrate the performance of our models on well-known real-world time series datasets, which can be further categorized into periodic time series (Ziyin et al., 2020; Fan et al., 2022) and long-term time series (Zeng et al., 2022).

To assess whether underlying signals are learned effectively, we formulate tasks as either interpolation or extrapolation tasks; that is, given a discrete measurements of the underlying signal, we train the proposed model and test it in either interpolation (imputation) or extrapolation (forecasting) settings. Unless otherwise specified, the input to the signal is either spatial coordinates $x$ or temporal coordinates $t$ for spatial or temporal signals, respectively (see Table 4 for the summary). We use MSE for the evaluation metric and conduct the experiments with three different random seeds and present their mean and standard deviation of evaluation metrics. More detailed descriptions of experiments and additional analyses such as ablation studies are listed in Appendix.

**Experimental Environments** We implement NeRT and baselines with PYTHON 3.9.7 and PYTORCH 1.13.0 (Paszke et al., 2019) that supports CUDA 11.6. The experiments are conducted on systems equipped with Intel Core-i9 CPUs and NVIDIA RTX A6000, A5000 GPUs. All implementations of our proposed method can be reproduced by referring to the attached README.md. To benefit the community, the code will be released online.

**Baselines for Comparison** As baseline models, we consider SIREN (Sitzmann et al., 2020), FFN (Tancik et al., 2020) and WIRE (Saragadam et al., 2023), the three representative INR models. For fair comparison, we set the model sizes to be the same. All models are trained with Adam (Kingma & Ba, 2014) with a learning rate of 0.001. In addition, we use eight existing time series models including Transformer-based and NODE-based models as non-INR baselines (cf. Appendix J). See the full description of the experimental setup in Appendix G.

### 4.1 SCIENTIFIC DATA – SIMULATED DATASETS

In the scientific domain, datasets often exhibit periodic characteristics, and analyzing these pattern is crucial in the processing of understanding the data. We train and evaluate NeRT and the other existing baselines using a two scientific dataset: harmonic oscillation and 2D-Helmholtz equations (McClenny & Braga-Neto, 2020). In all benchmark datasets, NeRT can represent the data more delicately compared to other models, and it demonstrates a strong capability to learn the inherent periodicity within the data.

#### 4.1.1 HARMONIC OSCILLATION

To analyze the performance of NeRT, we use the harmonic oscillation trajectory data. Simply reusing existing INRs, however, results in poor performance even for learning a simple and noise-less

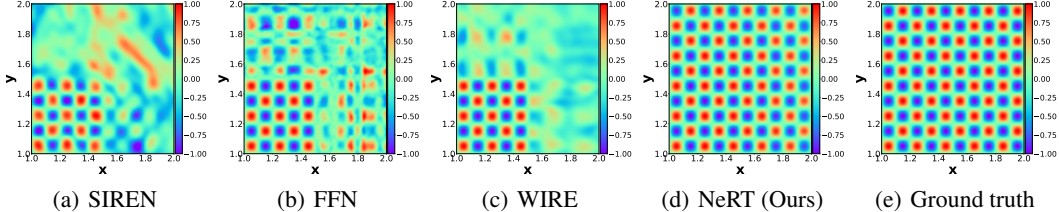

| (a) SIREN | (b) FFN | (c) WIRE | (d) NeRT (Ours) | (e) Ground truth |

**Figure 4: Extrapolation task on the 2D-Helmholtz equation.** Results of the extrapolation task with the training range of $x \in [1, 1.5]$ and $y \in [1, 1.5]$, i.e., the left-lower square, (Figures 4 (a)-(d)) and the ground-truth solution (Figure 4 (e)).

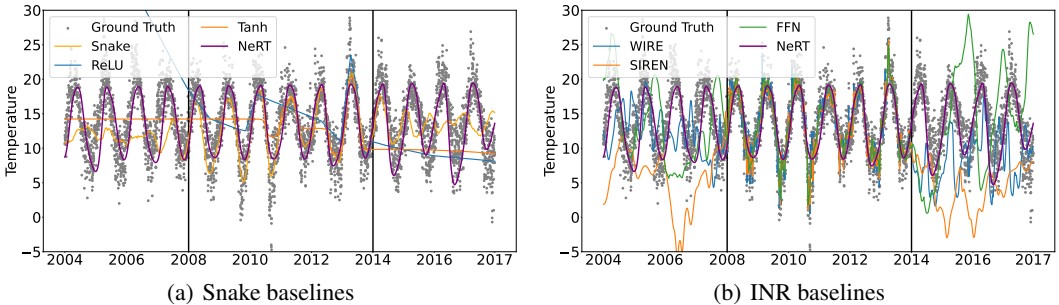

| (a) Snake baselines | (b) INR baselines |

**Figure 5: Comparison with Snake and INR baselines.** The middle area between the two solid vertical lines represents a training period.

simulated ODE data (cf. Figure 3). SIRENs, FFNs and WIREs are known to learn very complex (high-frequency) signals (e.g., images with details) by using either the sinusoidal activation, Fourier features or complex Gabor wavelet activation. However, as shown in Figure 3, these models perform poorly in the extrapolation task, while NeRT produces accurate predictions. (See Appendix E for more details on experimental setting.)

### 4.1.2 PDEs WITH PERIODICITY

Here, we demonstrate that NeRT can be applied another domain, namely learning solutions of PDEs — as a matter of fact, PDEs are implicit functions describing dynamics in the spatiotemporal coordinate system. Thus, we test NeRT on learning the solutions of 2D-Helmholtz equation, which is known to produce periodic behaviors. We refer readers to Appendix H for details.

**Experimental Setups** We perform extrapolation tasks over the spatial domain, where the model is trained by using a set of collocation points $(x, y) \in [1, 1.5]^2$ and is tested on $(x, y) \in [1, 2]^2 \backslash [1, 1.5]^2$. The 2D Helmholtz equations are used as a benchmark problem in McClenny & Braga-Neto (2020). The form of the PDE is as follows:

$$\frac{\partial^2 u(x,y)}{\partial x^2} + \frac{\partial^2 u(x,y)}{\partial y^2} + k^2 u(x,y) - q(x,y) = 0$$

$$q(x,y) = (-(a_1\pi)^2 - (a_2\pi)^2 + k^2)\sin(a_1\pi x)\sin(a_2\pi y)$$

(4)

where $k$ is a constant, and $q(x, y)$ is a source term. NeRT and baselines are trained to predict the solution $u$ at a given location $(x, y)$. Unlike the multi-variate time series, $u$ is uni-variate, so NeRT uses only $\psi_t$ as its encoder — we feed the raw coordinate $(x, y)$ instead of $\mathbf{c}_i^t$.

**Experimental Results** As shown in Figure 4, all three INR-based models have no difficulties in being overfitted to the training data. However, only the proposed method extrapolates accurately in the test region, while the other two baselines fail. We provide the analyses of additional experimental results with in Appendix H.

Table 1: **Experimental results on periodic time series.** The experimental results of the same model, presented on the same row, are from a single training, and the best results are reported in boldface. Errors measured in Mean Squared Error (MSE) are reported.

| Dataset | # blocks | Interpolation | | | | Extrapolation | | | |
|---|---|---|---|---|---|---|---|---|---|
| | | SIREN | FFN | WIRE | NeRT | SIREN | FFN | WIRE | NeRT |
| Electricity | 1 | 0.0200 | 0.0189 | 0.0170 | **0.0061** | 0.0256 | 0.0144 | 0.0244 | **0.0057** |
| | 2 | 0.0182 | 0.0144 | 0.0157 | **0.0057** | 0.0233 | 0.0142 | 0.0241 | **0.0077** |
| | 3 | 0.0183 | 0.0148 | 0.0168 | **0.0056** | 0.0231 | 0.0142 | 0.0251 | **0.0088** |
| Traffic | 1 | 0.0174 | 0.0140 | 0.0187 | **0.0057** | 0.0176 | 0.0113 | 0.0200 | **0.0050** |
| | 2 | 0.0169 | 0.0121 | 0.0191 | **0.0062** | 0.0181 | 0.0127 | 0.0199 | **0.0057** |
| | 3 | 0.0169 | 0.0114 | 0.0180 | **0.0055** | 0.0185 | 0.0130 | 0.0200 | **0.0115** |
| Caiso | 1 | 0.0230 | 0.0179 | 0.0182 | **0.0047** | 0.0596 | 0.0495 | 0.0466 | **0.0131** |
| | 2 | 0.0206 | 0.0179 | 0.0165 | **0.0041** | 0.0427 | 0.0399 | 0.0485 | **0.0148** |
| | 3 | 0.0337 | 0.0326 | 0.0270 | **0.0099** | 0.0260 | 0.0196 | 0.0473 | **0.0051** |
| NP | 1 | 0.0510 | 0.0502 | 0.0458 | **0.0149** | 0.0326 | 0.0346 | 0.0442 | **0.0235** |
| | 2 | 0.0464 | 0.0415 | 0.0424 | **0.0186** | 0.0329 | 0.0313 | 0.0443 | **0.0273** |
| | 3 | 0.0476 | 0.0431 | 0.0434 | **0.0196** | 0.0334 | 0.0304 | 0.0453 | **0.0274** |

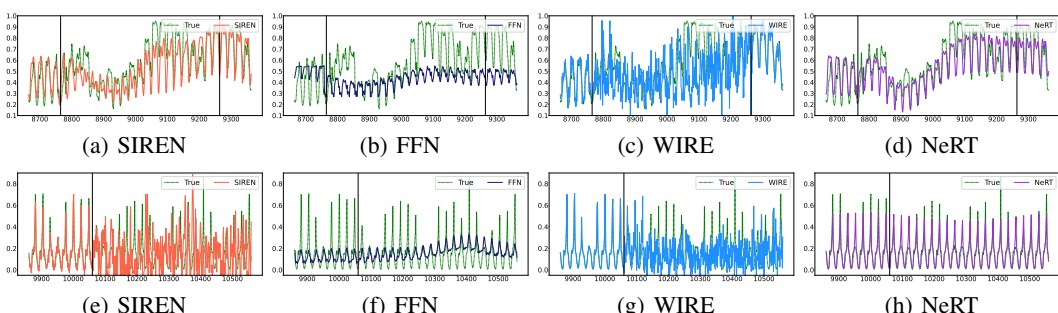

| (a) SIREN | (b) FFN | (c) WIRE | (d) NeRT |
|---|---|---|---|

| (e) SIREN | (f) FFN | (g) WIRE | (h) NeRT |
|---|---|---|---|

Figure 6: **Forecasting and imputation** [Top] Imputation results in NP ((a)-(d)), the middle area between the two vertical lines represents a testing block. [Bottom] Forecasting results in Traffic ((e)-(h)), where the area on the right of the vertical line represents a testing range.

## 4.2 REAL-WORLD TIME SERIES DATA

Now, we compare the performance of NeRT and baselines on real-world time series: Two periodic and long-term ones. Experimental results show that the proposed method outperforms in both scenarios, even for the long-term time series, which typically has much weaker periodicity than the periodic time series. On top of that, to better represent the real-world time series data, we suggest an additional module for encoding time series (cf. Appendix C); in short, this module leverages calendar representation of time indices, which are readily available in the dataset. In Sections 4.2.2–4.2.3, we apply the encoding module to every model, including baselines, and also provide ablation studies on the spatiotemporal coordinate.

### 4.2.1 ATMOSPHERIC DATA

**Experimental Setups** We use the atmospheric data from Minamitorishima Island as a benchmark dataset known to exhibit periodicity (Ziyin et al., 2020). The daily maximum temperature data from April 2008 to April 2014 is used as the training dataset, while the periods from April 2004 to April 2008 and from April 2014 to April 2017 are used for test. All other experimental settings follows the Ziyin et al. (2020).

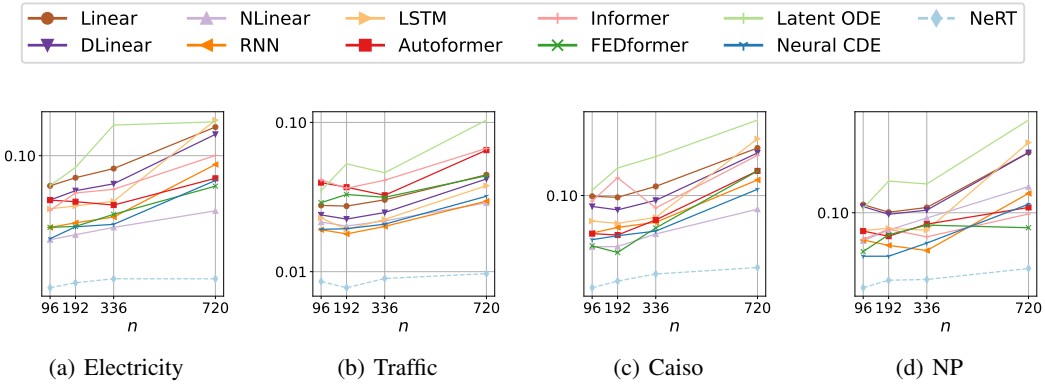

Figure 7: Comparisons with time series baselines for varying $n = \{96, 192, 336, 720\}$ (output/prediction window size) with a fixed input window size $m = 48$.

**Comparison with Existing Baselines**    To verify the performance of NeRT, we first construct INR baselines using SIREN, FFN and WIRE. Additionally, we utilize various activation functions used as baselines in Ziyin et al. (2020), including MLPs with ReLU, Tanh, and Snake.

As shown in Figure 5, MLPs with Snake, INR baselines and NeRT successfully represent the training period. However, baselines applying ReLU and Tanh activation functions in simple feedforward networks struggle to represent the data during the training interval. In the test period, all baselines, including Snake, SIREN, FFN and WIRE fail in predictions, while only NeRT successfully captures periodicity and predicts the test interval.

### 4.2.2   PERIODIC TIME SERIES

**Experimental Setups**    For periodic time series experiments, we select four uni-variate time series datasets, i.e., Electricity, Traffic, Caiso, and NP, which are all famous benchmark datasets used in Fan et al. (2022) and are known to have some periodic patterns. To demonstrate the efficacy on learning the periodic time series, we conduct interpolation and extrapolation tasks on missing blocks, each of which with a length of 500 observations — for hourly observations, 500 observations correspond to three weeks. We consider up to three missing blocks, i.e., 9 weeks. Since INR models are able to solve interpolation and extrapolation simultaneously, both tasks are tested by a single trained model. Detailed experimental settings such as the location of the blocks and hyperparameters are described in Appendix I.

**Comparison with the Existing INRs**    First we present the comparisons against the INR baselines, SIREN, FFN, and WIRE in Table 1 (see Table 11 for standard deviations of the metric), which essentially shows that NeRT outperforms in every task in all scenarios. Especially for Caiso, NeRT exhibits significantly lower errors both in the interpolation and extrapolation tasks, which are around a quarter of those of baselines. Notably, NeRT shows reasonably small MSEs even for extrapolating the last 1,500 observations. Figure 6 visualizes how three models interpolate in NP (Figures 6 (a)-(d)) and extrapolate in Traffic (Figures 6 (e)-(h)). In both cases, only NeRT shows good predictions while two other baselines struggle. An additional set of experiments with modulated versions of all INRs can be found in Appendix K.

**Comparison with Non-INR Baselines**    To provide a more comprehensive assessment, we conduct experiments comparing NeRT to eight existing time series models, including popular Transformer-based baselines. (1) Accuracy: In these experiments, we measure forecasting results by varying input/output window size, $m$ and $n$ and the full results are reported in Tables 12, 13 and 14 in Appendix. Figure 7 illustrates the changes in MSE with varying $n$ with $m = 48$. Notably, NeRT consistently achieves lower MSE compared to all other baseline models; particularly, the slope of NeRT's MSE is much lower than those of other methods' MSE. We emphasize that NeRT does not require retraining while the other methods need to be retrained for varying $n$. For detailed experimental setups and results, refer to Appendix J. (2) Computational/memory costs: Table 15 in

Table 3: Experimental results of ablation studies on Fourier feature mapping

| Dataset | NeRT (Fixed Fourier mapping) | | NeRT (Learnable Fourier mapping) | |
| --- | --- | --- | --- | --- |
| | Interpolation | Extrapolation | Interpolation | Extrapolation |
| Electricity | 0.0079±0.0015 | 0.0069±0.0009 | **0.0061**±**0.0006** | **0.0057**±**0.0007** |
| NP | 0.0190±0.0016 | 0.0313±0.0044 | **0.0149**±**0.0005** | **0.0235**±**0.0023** |

Appendix assures that NeRT has advantages in terms of the overall computational/memory efficiency than the baselines.

### 4.2.3 LONG-TERM TIME SERIES

**Experimental Setups**   We conduct experiments on general real-world long-term time series datasets to show the scalability of NeRT. The benchmark datasets of the long-term series forecasting task (Wen et al., 2022) are used for our experiments. Since the periodicity is typically weak in those datasets, we consider this task is much more challenging. We randomly drop 30%, 50%, and 70% observations and evaluate the interpolating performance using MSE as a metric.

**Experimental Results**   Due to space reasons, we list the experiments only with two datasets, ETTh1 and National Illness, in conjunction with their detailed experimental settings. Full results are in Appendix L. According to Table 2, in every drop ratio, NeRT beats other baselines by a large margin. For example, in ETTh1, NeRT shows an MSE of 0.0911, while SIREN and FFN shows 0.2173 and 0.3407, respectively. Moreover, in National Illness, NeRT outperforms other baselines by 489%.

Table 2: Long-term time series

| Dataset | Drop ratio | SIREN | FFN | NeRT |
| --- | --- | --- | --- | --- |
| | 30% | 0.1945 | 0.2522 | **0.0828** |
| ETTh1 | 50% | 0.2173 | 0.3407 | **0.0911** |
| | 70% | 0.2605 | 0.4256 | **0.1257** |
| | 30% | 0.3502 | 0.1110 | **0.0239** |
| National Illness | 50% | 0.1716 | 0.2319 | **0.0291** |
| | 70% | 0.3564 | 0.4453 | **0.0871** |

### 4.2.4 ABLATION STUDIES

**Ablation Studies on the Spatiotemporal Coordinate**   For all models (SIREN, FFN, and NeRT), we test the effect of the proposed coordinate system. In all three models, using the proposed coordinate system is critical in achieving better performance, which is reported in Appendix C.1.

**Ablation Studies on Learnable Fourier Feature Mapping**   To evaluate the effect of learnable Fourier feature mapping, we design experiments comparing its performance with a fixed Fourier feature mapping. The fixed Fourier feature mapping fixes the learnable parameters of the learnable Fourier feature mapping described in Equation 2, specifically, $\omega_m$, $\mathbf{A}_m$, $\mathbf{B}_m$, and $\delta_m$. The fixed values are set to be identical to the initial values of the learnable Fourier feature mapping, as discussed in Section 3.1. These experiments are conducted on the "one missing block" setting from Table 1. We use electricity and NP datasets for this evaluation, and the results are summarized in Table 3. As shown in Table 3, using learnable Fourier mapping in NeRT yields better performance compared to using fixed Fourier mapping in both interpolation and extrapolation tasks. That is, by adapting learnable factors to Fourier feature mapping, NeRT successfully learns a set of frequencies that better represents the specific time series.

### 4.3 DISCUSSION

Although the main goal here is to learn periodic signal effectively, the proposed approach can be also considered as one alternative method for time series modeling. The method has many unique and desirable aspects compared to traditional time series modeling approaches such as RNNs and their variants (e.g., LSTM) (Connor et al., 1994; Hochreiter & Schmidhuber, 1997; Qin et al., 2017; Lai et al., 2018; Sherstinsky, 2020; Hess et al., 2023; Koppe et al., 2019; Mikhaeil et al., 2022; Brenner et al., 2024), Transformers (Vaswani et al., 2017; Zhou et al., 2021; Wu et al., 2021; Liu et al., 2021; Wen et al., 2022; Zhou et al., 2022), Neural ODEs (Chen et al., 2018), and Neural CDEs (Kidger et al., 2020). As shown in the sets of experiments, the proposed algorithm is capable of efficiently handling

irregularly measured time series data, operating in a non-autoregressive way (both in training and test time), and forecasting based on the entire history (i.e., not relying on sliding windows). While promising, the proposed method needs improvements in various aspects to be considered as a serious competitor in the time series modeling business; for example, how to emphasize more on recent trend to make precise predictions, how to adjust model parameters efficiently based on new incoming observations, how to extract meta-knowledge of time series data from multiple time series instances of the same kind, etc. The concept of "modulation" and an accompanying meta-learning algorithm (as shown in Section K) offer preliminary insights into addressing these questions. However, a comprehensive exploration of these aspects is beyond the scope of this study. Our primary objective here is to effectively learn periodic signals by designing the new architecture for INRs.

## 5   RELATED WORK

**Implicit Neural Representations**   INR approaches, including physics-informed neural networks (PINN) (Raissi et al., 2019) for solving PDE problems and neural radiance fields (NeRF) (Mildenhall et al., 2021) for 3D representation, are quickly permeating various fields. In addition, a countermeasure to the spectral bias (Rahaman et al., 2019) in vanilla multi-layer perceptrons has been proposed, enabling more sophisticated INR-based representations (Sitzmann et al., 2020; Saragadam et al., 2023). According to Tancik et al. (2020), it is shown that the infinitely wide ReLU based neural network with random Fourier features are equivalent to the shift-invariant kernel method in the perspective of NTK (Jacot et al., 2018). Further extended from Fourier-mapped input networks, the work in Benbarka et al. (2022) draws a connection to Fourier series, leading to MLPs with an integer lattice mapping input. In the time series domain, there exist INR-based studies on unsupervised anomaly detection (Jeong & Shin, 2022) and time series forecasting (Woo et al., 2022). Relevant works in computer vision domain includes NPP (Chen et al., 2022), which learns periodic signal by developing a periodicity detector and patch-wise matching.

**Decomposition Methods Used in Time Series Modeling**   Decomposition methods aim to separate a time series sample into multiple components, often including a trend, seasonality, and residual component (Cleveland et al., 1990). These components can then be modeled separately, allowing for more accurate predictions and insights. There exist various decomposition-based methods that have been used for time series, ranging from traditional time series models to recent deep learning models. For example, the wavelet decomposition decomposes time series into different frequency bands (Percival & Walden, 2000; Wang et al., 2018) and the singular spectrum analysis (SSA) decomposes time series into a set of eigenvectors to extract various oscillatory patterns which are interpretable (Vautard & Ghil, 1989; Sulandari et al., 2020).

## 6   CONCLUSIONS

Due to the strength in learning coordinate-based systems, INR has high potential for various fields in natural sciences and engineering. However, despite the promising nature of INR, it has been rarely applied to functions with periodic behavior (e.g., time series), and no existing unified models are based on the INR paradigm. Therefore, we aim to address the limitations that conventional time series models possess and propose NeRT which resorts to the advantages of INR to effectively resolve existing challenges in the field. Based on the INR framework, NeRT effectively learns and predicts sequential data by separating the periodic and scale factors. Additionally, we suggest a method for embedding the spatiotemporal coordinate of multivariate time series. Through this approach, NeRT clearly outperforms existing INR models and can represent the sequential data more accurately.

**Limitations and Future Works**   Compared to existing time series models which rely on a concept of sliding-windows, INR-based modeling approaches proposed in this paper requires only coordinate-based queries for predictions during inference. This is extremely computationally efficient, but at the same time could cause a lack of adaptivity when a new set of data is available to fine-tune the model. To mitigate this issue, we adopt the idea of *modulation* from Dupont et al. (2022), which is shown to be effective in resolving such issue to some extent. Enabling the model to adapt to *shift modulation* can be challenging even with the latent modulation; however, we consider this as the future direction for this research and we focus on learning functions with periodic behaviors in this study.

## 7 ACKNOWLEDGEMENT

This work was supported by Institute of Information & communications Technology Planning & Evaluation (IITP) grant funded by the Korea government(MSIT) (No. RS-2024-00457882, AI Research Hub Project). K. Lee acknowledges support from the U.S. National Science Foundation under grant IIS 2338909. K. Lee also acknowledges Research Computing at Arizona State University for providing HPC resources that have contributed to the partial research results reported within this paper.

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

# A  SUMMARY OF DATASETS

Table 4: Summary of datasets used in the experiments

|  | Problem | Input | Physical meaning |
|---|---|---|---|
| Toy example (Section 1) | $\sin(50x)$ | $x$ | 1D spatial coordinate |
| Scientific data (Section 4.1) | Harmonic oscillator
2D Helmholtz equations | $t$
$x$ | temporal coordinate
2D spatial coordinate |
| Time series data (Section 4.2) | Atmospheric data
Periodic time series data
Long-term time series data | $t$
$t$
calendar information | temporal coordinate
temporal coordinate
temporal coordinate |

The summary of the datasets used in our experiments is presented in Table 4. We employ a sinusoidal wave as a toy example and two ideal scientific datasets obtained from ODE and PDE as benchmark datasets. Additionally, we evaluate NeRT's performance on real-world time series data, using a total of seven real-world benchmark datasets: four types of periodic time series datasets and three types of long-term time series datasets. In every dataset, NeRT outperforms other baselines.

# B  COMPARISONS TO THE EXISTING TIME SERIES MODELING APPROACH

Although the proposed method mainly focuses on learning periodic signals, it is highly relevant to time series modeling and we discuss the proposed method as an alternative model for time series models.

As time series processing has been one fundamental task of machine learning, many different deep-learning (DL) algorithms have been studied. Recurrent neural networks (RNNs) and their variants (e.g., long short-term memory, or LSTM) (Connor et al., 1994; Hochreiter & Schmidhuber, 1997; Qin et al., 2017; Lai et al., 2018; Sherstinsky, 2020) have been (one of) the first DL algorithms for processing time series data. Transformers (Vaswani et al., 2017; Zhou et al., 2021; Wu et al., 2021; Liu et al., 2021; Wen et al., 2022; Zhou et al., 2022) quickly superseded RNNs thanks to its higher representation learning capability aided by the self-attention. Another line of modeling approaches include the differential equation-based DL paradigm, also known as *continuous-time* methods, which exhibit favorable features for handling irregularly-sampled time series neural ordinary differential equations (NODEs) (Chen et al., 2018) and neural controlled differential equations (NCDEs) (Kidger et al., 2020) are two exemplary works in this line of research.

INR-based time series modeling as proposed in the current manuscript exhibits some advantages over previous time series modeling approaches. First, the proposed method is naturally capable of processing irregular time series, which is often challenging to typical time series models. Some remedies include time series embedding (Kazemi et al., 2019), positional encoding (Vaswani et al., 2017), padding, or likelihood-based approaches (Mei & Eisner, 2017), which partially resolves the difficulty at increased cost. Secondly, the proposed method operates without the concept of a sliding window (i.e., reading $m$ past observations and predicting $n$ observations ahead, where $m$ and $n$ are hyperparameters). The performance of traditional approaches relying on the sliding window is often highly sensitive to the hyperparameters (cf. Appendix B.1). Lastly, the proposed method can be trained even in the case where only a single time series instance is available.

The weaknesses or the limitation of the proposed method are highly related to some items of the listed advantages. The vanilla INR-based modeling does not provide a means to place more emphasis on recent measurements for forecasting (e.g., no sliding windows), which is a typical practice in other time series modeling approaches. Also, it typically trains a single INR for an individual time series instance, making it challenging to extract features that are common to a dataset level.

As mentioned above, to address such weaknesses, the concept of "modulation" and an accompanying meta-learning algorithm (as shown in Section K) can be utilized. However, it requires further significant investigations, which has not been pursued in this study as the main goal of this study is to learn periodic signal effectively with the new architecture design.

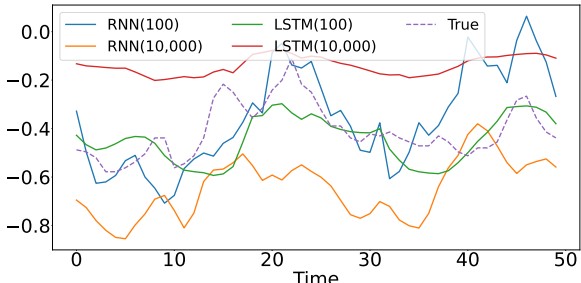

Table 5: Experimental results of time series models on varying window-sizes

| Model | 100 | 10000 |
|-------|------|-------|
| RNN | 0.5999 | 0.4478 |
| LSTM | 0.6927 | 1.0530 |

Figure 8: Visualization of results according to window-size of existing time series models

### B.1 SENSITIVITY OF WINDOW-BASED TIME SERIES MODELS

The commonly used approach in time series models, called *shifting window*, assumes a fixed window size. However, this method has several critical drawbacks. First, it requires finding an optimal input window size. Using a too small window size may result in small and inference time, but the model sees short patterns and struggles to infer effectively. On the other hand, employing a too large input window size can enable capturing long-term patterns but at the same time can lead to long training and inference time. Thus, the window size is a highly sensitive hyperparameter, and finding a balance between these two settings is a challenging task.

To empirically show how the size of window affects the model in forecasting time series, we compare results of forecasting 50 future observations by varying the input window size in $\{100, 10000\}$ with other conditions fixed. For the experiment, we choose two typical time series models, RNN and LSTM, and train them for 1,000 epochs on ETTh1. As shown in Table 5, for both RNN and LSTM, their model predictions show big differences depending on the window size, in terms of MSE. Note that models with a small window size, i.e., a window size of 100, shows better predictions when using RNN, while LSTM shows lower MSE when using a longer window size, i.e., a window size of 10000. The same trend can also be observed in Figure 8. In Figure 8, predicted values exhibit significant differences. Therefore, the window size is a critical hyperparameter in modeling time series.

## C SPATIO-TEMPORAL COORDINATE SYSTEMS OF TIME SERIES DATA

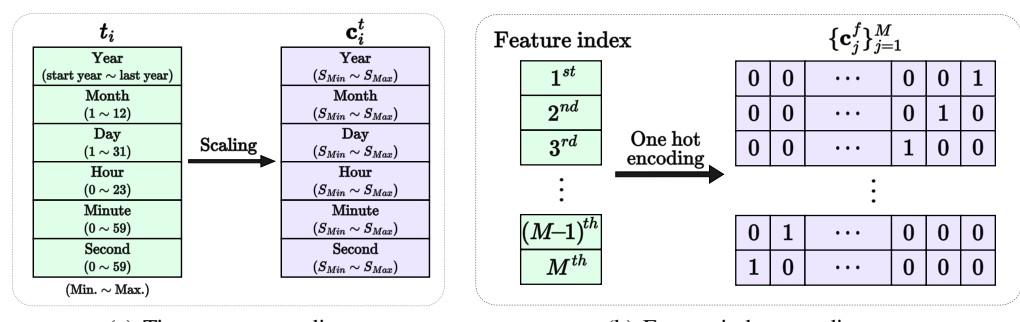

(a) Time stamp encoding

(b) Feature index encoding

Figure 9: **Spatio-temporal coordinate construction.** Our method to define temporal coordinates $\{\mathbf{c}_i^t\}_{i=1}^N$ (Figure 9 (a)) and spatial coordinates $\{\mathbf{c}_j^f\}_{j=1}^M$ (Figure 9 (b)).

Unlike domains of applications where the coordinate system is relatively well-defined (i.e., 2d Cartesian coordinate systems for images or solutions of partial differential equations, PDEs), multi-variate time series modeling needs a new definition of a coordinate system. Given an $M$-dimensional multi-variate time series $\{\mathbf{x}_i\}_{i=1}^N$, where $\mathbf{x}_i = \mathbf{x}(t_i) \in \Omega^M$, a naïve way of building INRs is to

directly use the time stamps, $t_i$, as a coordinate and $\mathbf{x}_i$ as a signal intensity. However, such a coordinate system is too primitive to properly represent multi-variate time series data in INRs.

Instead, we propose to manufacture a refined coordinate system, providing fine-granularity in indexing temporal domain and feature domain (See Figure 9). We interpret the feature domain, i.e., the $M$-dimensional space, $\Omega^M$, as a spatial domain and, thus, propose a novel spatio-temporal coordinate system for multi-variate time series data. In our new proposed coordinate system, a coordinate can be expressed as $\{\mathbf{c}_i := \mathbf{c}_i^t, (\{\mathbf{c}_j^f\}_{j=1}^M)\}_{i=1}^N$, where $\mathbf{c}_i^t \in \mathbb{R}^{D_{\mathbf{c}^t}}$ and $\mathbf{c}_j^f \in \mathbb{R}^{D_{\mathbf{c}^f}}$. Here, $\mathbf{c}_i^t$ and $\mathbf{c}_j^f$ denote the temporal and the spatial coordinates of $t_i$ ($i \in N$) and the $j$-th feature ($j \in M$), respectively.

**Spatial coordinates** For the spatial coordinate, we employ one-hot encoding, which translate an integer index into one-hot vectors. Unlike images or PDE solutions, the spatial locality in the feature domain is less clear. Moreover, imposing a certain order based on the integer index is likely to impose undesirable biases. Thus, we propose to use one-hot encoding, which removes reliance on a specific ordering and is general enough to represent represent many different multi-variate time series.

**Temporal coordinates** For the temporal coordinate, when the calendar information is readily available (as in Section 4.2.3), we fully utilize the information; we apply a simple min-max scaling to put different quantitative values (year, month, day, etc) into the same numerical scale $[S_{\min}, S_{\max}]$, effectively resolving numerical issues.

### C.1 ABLATION STUDIES ON THE SPATIO-TEMPORAL COORDINATE

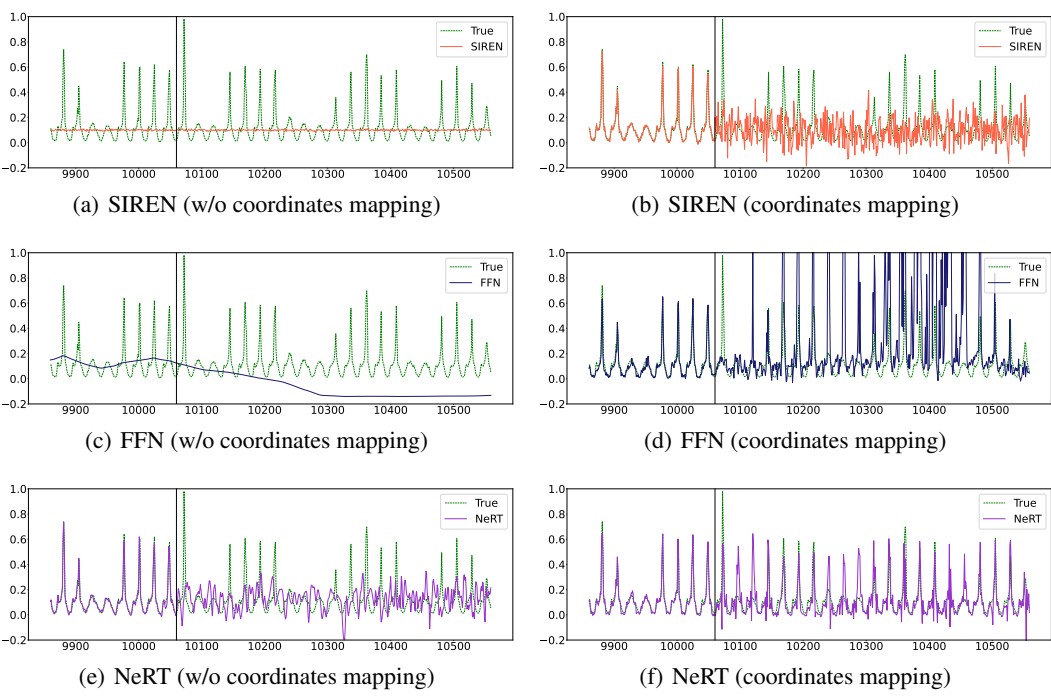

Figure 10: Experimental results of ablation study according to the spatiotemporal coordinates mapping introduced in Appendix C. The figures in left column are the results without coordinates mapping and the figures in right column are the results using coordinates mapping.

As an ablation study, we aim to investigate the impact of the proposed coordinates mapping in Figure 9 on INR models. The overall experimental setup follows that of section 4.2. To ensure a fair comparison, we set the model size to be the same across all models. In addition, the initial frequency $w$ of SIREN and NeRT is fixed at 30, which is known to work well in Sitzmann et al. (2020). The results presented in Figure 10(a), (c), (e) correspond to the models trained without a coordinates mapping, where a one-dimensional scaled time-stamp used as input to the model. For the baselines,

SIREN and FFN, it can be observed that they struggle to accurately represent the overall domain, while NeRT exhibits a more refined representation. However, NeRT still exhibits discrepancies from the ground truth in extrapolation. As shown in Figure 10(b), (d), (f), which is the result of using coordinates mapping, it can be observed that INR models can depict time series more accurately when input coordinates are mapped. In particular, NeRT can even perform sophisticated extrapolation inference.

# D   ANALYSES ON NERT

## D.1   FOURIER FEATURE MAPPING IN TEMPORAL COORDINATE TRANSFORMATION

The Fourier feature mapping introduced in Tancik et al. (2020) transforms the input coordinates using periodic functions, allowing the neural networks to solve the spectral bias in MLPs. This approach is a simple yet powerful way to address the problem. Moreover, the Fourier feature mapping exhibits shift-invariant properties from an NTK perspective. Our learnable Fourier feature mapping enjoys all the advantages of the Fourier feature mapping and moreover, it addresses the difficulty of finding a task-specific fixed set of frequencies in the Fourier feature mapping since we learn them as well. As explained in Section 2, NeRT maps the temporal coordinate onto a desired closed finite interval $[S_{min}, S_{max}]$. Therefore, NeRT can approximate discrete coordinate-based time series as continuous function in the closed domain and thus, the extrapolation in the original temporal coordinate can be somehow considered as an interpolation in the learned coordinate, e.g., everyday 12pm has $S_{min}$ regardless of year and month. Under these conditions, when coordinates outside the training range are inputted, the domain of NeRT is converted to the maximum and minimum values of $\psi_F$ by the extreme value theorem. In other words, by Equation 2, NeRT can operate in a wide range of the temporal coordinate, and this approach works very effectively, especially when performing extrapolations.

# E   ADDITIONAL EXPERIMENTS WITH AN ODE-BASED SYNTHETIC TIME SERIES

## E.1   EXPERIMENTAL SETUPS

To evaluate the effectiveness of our proposed NeRT in an ideal setting without any noise, we conduct experiments (cf. Figure 3). We use the damped oscillation ODE, which can represent harmonic and oscillatory motion, as the benchmark dataset. The specific equation and analytic solution are as follows:

$$m_d\frac{d^2x}{dt^2} + b_d\frac{dx}{dt} + k_dx = 0,$$
$$x(t) = A_d e^{-\frac{b_d}{2m}t}\cos(\omega_d t + \varphi). \tag{5}$$

Equation 5 represents the motion equation that accounts for all the forces acting on the object, where $m_d$ denotes the mass of an object, and $b$ and $k_d$ are physical constants. We set $m_d = 1$, $b_d = 0$ (undamping) or 4 (damping), $\omega_d = 50$, and $A_d = 10$. All experiments employ 2,000 epochs.

## E.2   $n^{th}$ ORDER DERIVATIVE PENALTY

We propose an additional penalty term for use in the training process of INR to represent smooth dynamics and prevent overfitting. Since NeRT directly takes coordinates as input, it can compute the $n^{th}$ partial derivative of the output with respect to the input coordinates. In the harmonic oscillation experiment, we aim to refine the periodicity/scale by adding the third-order partial derivative term with respect to time coordinate of the output to the loss.

## E.3   EXPERIMENTAL RESULTS

In this section, we do additional experiments on undamping/damping oscillatory signals. For each ODEs, we do interpolation, extrapolation, and a mixed task, which do both interpolation and

extrapolation at the same time. Since extrapolation task for a damping oscillatory signal is in the main paper, we list results for other five experiments, and the results are summarized in Table 6.

Table 6: Additional results with an ODE-based synthetic time series

|  | Task | SIREN | FFN | WIRE | NeRT |
|---|---|---|---|---|---|
| Undamping | Interpolation | 96.6879$\pm$31.0011 | 0.0856$\pm$0.0445 | 55.0295$\pm$2.4795 | **0.0183$\pm$0.0123** |
|  | Extrapolation | 121.8907$\pm$23.9423 | 1.0917$\pm$0.7321 | 52.8508$\pm$2.1014 | **0.4109$\pm$0.4335** |
|  | Interp+Extrap | 93.3860$\pm$38.3511 | 0.1270$\pm$0.0888 | 50.6520$\pm$5.8592 | **0.0336$\pm$0.0204** |
| Damping | Interpolation | 0.1846$\pm$0.0796 | 0.0017$\pm$0.0006 | 0.2258$\pm$0.0218 | **0.0011$\pm$0.0008** |
|  | Interp+Extrap | 0.2758$\pm$0.1007 | 0.0034$\pm$0.0018 | 0.1967$\pm$0.0093 | **0.0004$\pm$0.0002** |

As shown in Table 6, NeRT outperforms SIREN, FFN and WIRE in every task by a big margin, and visualizations of the results are summarized in Figures 11, 12, 13, 14, and 15.

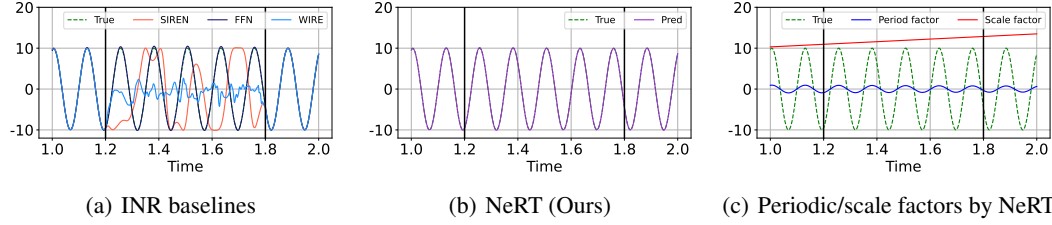

(a) INR baselines      (b) NeRT (Ours)      (c) Periodic/scale factors by NeRT

Figure 11: **Interpolation task on an undamping oscillatory signal.** Results of interpolation task with an ODE (Figures 11(a)-(b)) and extracted factors during training (Figure 11(c)). The inside of the two solid lines is a testing range ($t \in [1.2, 1.8]$) and the outside is a training range ($t \in [1.0, 1.2], [1.8, 2.0]$).

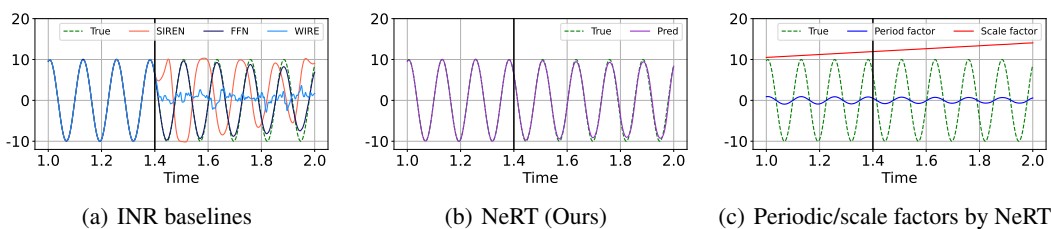

(a) INR baselines      (b) NeRT (Ours)      (c) Periodic/scale factors by NeRT

Figure 12: **Extrapolation task on an undamping oscillatory signal.** Results of extrapolation task with an ODE (Figures 12(a)-(b)) and extracted factors during training (Figure 12(c)). The left side of the solid line represents the training range ($t \in [1.0, 1.4]$), while the right side represents the testing range ($t \in [1.4, 2.0]$).

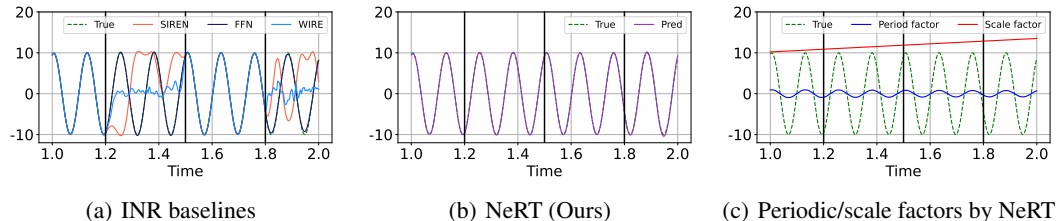

(a) INR baselines      (b) NeRT (Ours)      (c) Periodic/scale factors by NeRT

Figure 13: **Interpolation and Extrapolation task on an undamping oscillatory signal.** Results of interpolation and extrapolation tasks with an ODE (Figures 13(a)-(b)) and extracted factors during training (Figure 13(c)). The training range are $t \in [1.0, 1.2], [1.5, 1.8]$, and the testing range are $t \in [1.2, 1.5], [1.8, 2.0]$.

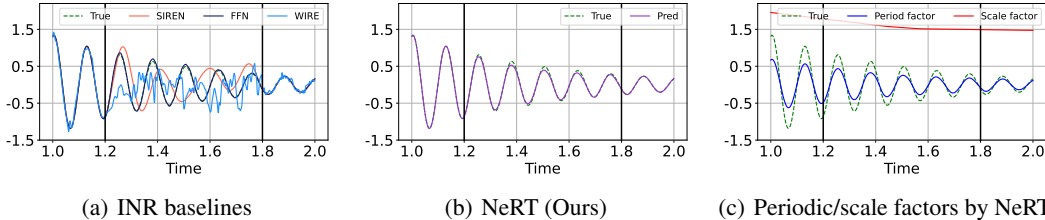

|(a) INR baselines|(b) NeRT (Ours)|(c) Periodic/scale factors by NeRT|

Figure 14: **Preliminary study with a damping oscillatory signal.** Results of interpolation task with an ODE (Figures 14(a)-(b)) and extracted factors during training (Figure 14(c)). The inside of the two solid lines represents the testing range ($t \in [1.2, 1.8]$, while the outside represents the training range ($t \in [1.0, 1.2], [1.8, 2.0]$).

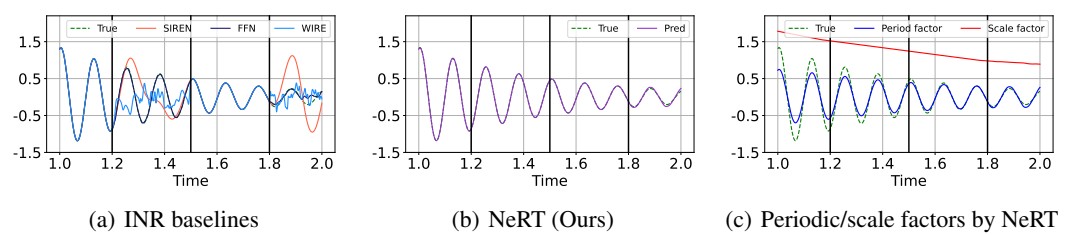

|(a) INR baselines|(b) NeRT (Ours)|(c) Periodic/scale factors by NeRT|

Figure 15: **Preliminary study with a damping oscillatory signal.** Results of interpolation and extrapolation tasks with an ODE (Figures 15(a)-(b)) and extracted factors during training (Figure 15(c)). The training range are $t \in [1.0, 1.2], [1.5, 1.8]$, and the testing range are $t \in [1.2, 1.5], [1.8, 2.0]$.

## F  ALGORITHM

To provide detailed explanations of the training process of NeRT, we present the following training Algorithm 1.

---
**Algorithm 1** Training of the proposed method
---
/* Training */
**Training datasets:** $M$-variate sequence $\{(\mathbf{x}_i, t_i)\}_{i=1}^N$
**Input:** A set of training sampled coordinate: $\{\mathbf{c}_i^t, (\{\mathbf{c}_j^f\}_{j=1}^M)\}_{i=1}^N$
Initialize the parameters of NeRT $\{\theta_t, \theta_F, \theta_f, \theta_s, \theta_p\}$
**for** $epoch = 1$ to $ep$ **do**
    Compute forward pass: $\hat{x}_{i,j}((\mathbf{c}_i^t, \mathbf{c}_j^f); \{\theta_t, \theta_F, \theta_f, \theta_s, \theta_p\})$
    Compute MSE: $(\hat{x}_{i,j} - x_{i,j})^2$
    Compute backward pass and update the parameters
    Compute the validation error with validation data
**end for**
**Output:** Optimal parameters of NeRT (Lowest validation error)

---

# G    DETAILED DESCRIPTION OF DATASETS

## G.1    2D-HELMHOLTZ EQUATION

The 2D Helmholtz equation, used in Section 4.1, is a differential equation utilized for modeling wave and electromagnetic phenomena. We experiment with the condition of the Equation 4 and are able to directly obtain the analytical solution $u(x, y) = \sin(a_1 \pi x) \sin(a_2 \pi y)$.

## G.2    TIME SERIES DATASETS

Table 7: **Dataset statistics.** Max. length (resp. Min. length) is the longest (resp. shortest) timestamp length among the timestamps of the features in the samples.

| Dataset | Periodic time series | | | | Long-term time series | | |
|---|---|---|---|---|---|---|---|
| | Electricity | Traffic | Caiso | NP | ETTh1 | ETTh2 | National Illness |
| Frequency | hourly | hourly | hourly | hourly | hourly | hourly | weekly |
| Start date | 2012-01-01 | 2008-01-02 | 2013-01-01 | 2013-01-01 | 2016-07-01 | 2016-07-01 | 2002-01-01 |
| End date | 2015-01-01 | 2009-03-31 | 2021-06-30 | 2020-12-31 | 2018-06-26 | 2018-06-26 | 2020-06-30 |
| # features | 1 | 1 | 1 | 1 | 7 | 7 | 7 |
| Max. length | 26,304 | 10,560 | 74,079 | 70,120 | 17,420 | 17,420 | 966 |
| Min. length | 26,271 | 10,512 | 41,547 | 70,076 | 17,420 | 17,420 | 966 |

Experiments on time series datasets consists of two parts: i) periodic time series, and ii) long-term time series. We use four datasets used as benchmark datasets in Fan et al. (2022) for the periodic time series task and three datasets from Wen et al. (2022) for the long-term time series task. As shown in Table 7, in order to show how NeRT predicts in various scenarios, we choose datasets with a wide range of length and frequency, including both uni-variate and multi-variate datasets. Detailed descriptions of datasets are as follows:

- Periodic time series
    - Electricity contains hourly records of electricity consumption from 2012 to 2014.
    - Traffic consists of hourly data from the sensors in San Francisco freeways, providing information on the road occupancy rates between 2015 and 2016.
    - Caiso comprises hourly actual electricity load series in various zones across California from 2013 to 2021.
    - NP contains a collection of hourly energy production volume from 2013 to 2020 in several European countries.
- Long-term time series
    - ETTh1 and ETTh2 (Zhou et al., 2021) are hourly collected ETT (Electricity Transformer Temperature) datasets from July 2016 to July 2018.
    - National Illness is a weekly collected medical dataset from the Centers for Disease Control and Prevention of the United States. It contains the information of patients with influenza-like illness spanning from 2002 to 2021.

## H DETAILED EXPERIMENTAL RESULTS ON 2D HELMHOLTZ EQUATION

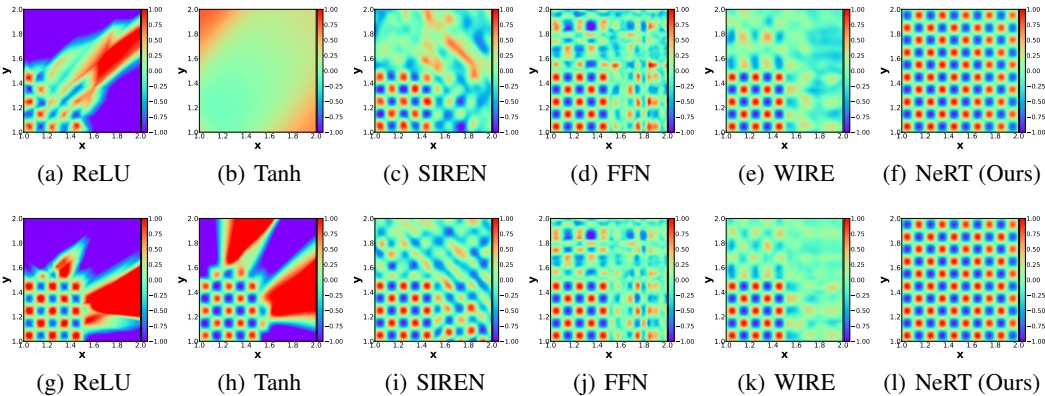

| (a) ReLU | (b) Tanh | (c) SIREN | (d) FFN | (e) WIRE | (f) NeRT (Ours) |
|:--:|:--:|:--:|:--:|:--:|:--:|
| (g) ReLU | (h) Tanh | (i) SIREN | (j) FFN | (k) WIRE | (l) NeRT (Ours) |

Figure 16: Detailed experimental results

Table 8: Experimental results of Helmholtz equations

| Epoch | ReLU | Tanh | SIREN | FFN | WIRE | NeRT |
|---|---|---|---|---|---|---|
| 1,000 | 3.7097±0.3393 | 0.3170±0.0406 | 0.3325±0.1177 | 0.4625±0.0820 | 0.2753±0.0270 | **0.0025±0.0006** |
| 10,000 | 20.2617±14.1040 | 2.4257±1.9618 | 0.2504±0.0356 | 0.4455±0.0769 | 0.2540±0.0232 | **0.0009±0.0005** |

In the extension of Section 4.1, to provide more comprehensive analysis, we increase the number of epochs and add a baseline. Figures 16(a)-(f) represent the results after training for 1,000 epochs, while Figures 16(g)-(l) depict the results after training for 10,000 epochs. Furthermore, keeping all hyperparameters and model sizes the same, we add baselines by changing the activation function to ReLU and Tanh. This is depicted in the first and second columns of Figure 16. As shown in Figures 16(a) and (b), models using ReLU and Tanh activation functions struggle even to learn the training range. On the other hand, all INR models demonstrate successfully precise learning of the training range ($x \in [1.0, 1.5], y \in [1.0, 1.5]$) within just 1,000 epoch. When training is extended to 10,000 epochs, all models managed to learn the training range, but only NeRT successfully predict the test range ($x \in [1.5, 2.0], y \in [1.5, 2.0]$).

# I  EXPERIMENTS ON PERIODIC TIME SERIES

## I.1  DETAILED EXPERIMENTAL SETUPS

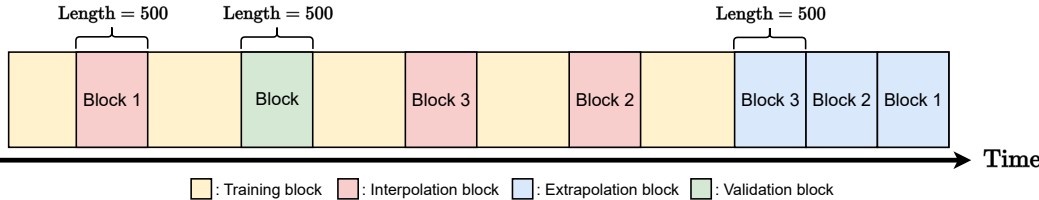

Figure 17: Experimental setup on periodic time series.

Table 9: Best hyperparameter configurations in the periodic time series task

|  | $\omega^{\text{init}}$ | $\omega^{\text{inner}}$ | $\dim(\psi_t)$ | $\dim(\psi_F)$ | $\dim(\mathbf{h}_p)$ | $\dim(\mathbf{h}_s)$ |
|---|---|---|---|---|---|---|
| Electricity | 5.0 | 1.0 | 30 | 30 | 10 | 30 |
| Traffic | 10.0 | 1.0 | 30 | 10 | 50 | 10 |
| Caiso | 10.0 | 1.0 | 10 | 30 | 50 | 10 |
| NP | 3.0 | 3.0 | 30 | 30 | 10 | 30 |

We design the experiment using the first 10 samples, each of which consists of 12 blocks (cf. Figure 17), from each dataset and conduct an experiment to fill in the values of missing blocks. In a sample, we perform both the interpolation and the extrapolation tasks. Detailed locations and constructions are summarized in Figure 17. As shown in Figure 17, in a sample there are three interpolation blocks colored in red, three extrapolation blocks colored in blue, one validation block colored in green, and the remaining yellow parts represent the training dataset. Each block has a length of 500, and the missing blocks for each task, as specified as "# blocks" in Table 1, are as follows:

- # blocks = 1 means that we perform the interpolation and extrapolation tasks for Block 1.
- # blocks = 2 means that we perform the interpolation and extrapolation tasks for Block 1 and Block 2.
- # blocks = 3 means that we perform the interpolation and extrapolation tasks for Block 1, Block 2 and Block 3.

A validation block is not used for training in every task, and used for the purpose of validation. Each model is trained for 2,000 epochs.

For hyperparameters, we set $S_{max}$ to 1 and for the fair comparison, we use similar model sizes for all methods and share the frequencies across the models. There are two frequencies used as hyperparameters, $\omega^{\text{init}}$ and $\omega^{\text{inner}}$. $\omega^{\text{init}}$ is used in our learnable Fourier feature mapping and corresponds to $b$ in Equation 2. $\omega^{\text{inner}}$ denotes the frequency of the sinusoidal function $\rho_s$ in Equation 3. For the number of layers, we set $L_t, L_f$ and $L_s$ to 2, and $L_p$ to 5. The best hyperparameter configurations of NeRT in the periodic time series task are summarized in Table 9. We use an additional FC layer after Equation 2 and let $\mathbf{h}_p$ and $\mathbf{h}_s$ be the hidden vectors of the period and scale decoders, respectively.

## I.2  ADDITIONAL EXPERIMENTAL RESULTS

We conduct periodic time series experiments on two numerical methods, Linear (linear interpolation) and Cubic (cubic interpolation), and report the results in Table 10. As shown in Table 10, NeRT beats two numerical methods. We compare NeRT to Linear and Cubic only for the interpolation task, since those numerical methods are not able to extrapolate.

Additionally, in Figures 18, 19, 20, and  21, we show the visualizations of the remaining datasets' interpolation and extrapolation results that are not included in the main paper. In Figures 18 and 20, we show the results at the best epoch, i.e., the lowest validation error, and in Figures 19 and 21, we show results at the last epoch, i.e., after 2,000 epochs. NeRT avoids overfitting to training data in both cases while other two baselines are significantly overfitted to training data and fail in the testing range.

In Figure 22, we propose how the periodic and scale factors of NeRT work in the periodic time series interpolation and extrapolation tasks. Figures 22 (a)-(d) correspond to the interpolation task and Figures 22 (e)-(h) correspond to the extrapolation task.

Table 10: **Additional experimental results on periodic time series.** The best results are reported in boldface.

| Dataset | # blocks | Interpolation | | |
| | | Linear | Cubic | NeRT |
|---|---|---|---|---|
| Electricity | 1 | 0.0126 | 61.4654 | **0.0061**±**0.0006** |
| | 2 | 0.0155 | 47.4215 | **0.0057**±**0.0005** |
| | 3 | 0.0182 | 41.9424 | **0.0056**±**0.0006** |
| Traffic | 1 | 0.0220 | 4.4286 | **0.0057**±**0.0002** |
| | 2 | 0.0191 | 3.0941 | **0.0062**±**0.0000** |
| | 3 | 0.0191 | 4.1447 | **0.0055**±**0.0002** |
| Caiso | 1 | 0.0223 | 8.8545 | **0.0047**±**0.0012** |
| | 2 | 0.0171 | 6.3525 | **0.0041**±**0.0002** |
| | 3 | 0.0171 | 6.4894 | **0.0099**±**0.0007** |
| NP | 1 | 0.0426 | 1.3798 | **0.0149**±**0.0005** |
| | 2 | 0.0477 | 2.2489 | **0.0186**±**0.0008** |
| | 3 | 0.0454 | 3.8557 | **0.0196**±**0.0006** |

Table 11: **Standard deviations of experimental results on periodic time series.** Each experiment is repeated three times.

| Dataset | # blocks | Interpolation | | | | Extrapolation | | | |
| | | SIREN | FFN | WIRE | NeRT | SIREN | FFN | WIRE | NeRT |
|---|---|---|---|---|---|---|---|---|---|
| Electricity | 1 | ±0.0007 | ±0.0021 | ±0.0014 | ±0.0006 | ±0.0005 | ±0.0009 | ±0.0004 | ±0.0007 |
| | 2 | ±0.0009 | ±0.0012 | ±0.0018 | ±0.0005 | ±0.0006 | ±0.0006 | ±0.0027 | ±0.0019 |
| | 3 | ±0.0004 | ±0.0013 | ±0.0017 | ±0.0006 | ±0.0001 | ±0.0005 | ±0.0020 | ±0.0008 |
| Traffic | 1 | ±0.0007 | ±0.0007 | ±0.0011 | ±0.0002 | ±0.0009 | ±0.0007 | ±0.0003 | ±0.0002 |
| | 2 | ±0.0003 | ±0.0001 | ±0.0010 | ±0.0000 | ±0.0004 | ±0.0003 | ±0.0004 | ±0.0003 |
| | 3 | ±0.0006 | ±0.0001 | ±0.0014 | ±0.0002 | ±0.0006 | ±0.0002 | ±0.0003 | ±0.0026 |
| Caiso | 1 | ±0.0008 | ±0.0013 | ±0.0016 | ±0.0012 | ±0.0018 | ±0.0045 | ±0.0010 | ±0.0014 |
| | 2 | ±0.0002 | ±0.0012 | ±0.0019 | ±0.0002 | ±0.0005 | ±0.0017 | ±0.0029 | ±0.0066 |
| | 3 | ±0.0004 | ±0.0017 | ±0.0016 | ±0.0007 | ±0.0007 | ±0.0008 | ±0.0022 | ±0.0008 |
| NP | 1 | ±0.0005 | ±0.0013 | ±0.0019 | ±0.0005 | ±0.0014 | ±0.0021 | ±0.0033 | ±0.0023 |
| | 2 | ±0.0004 | ±0.0002 | ±0.0019 | ±0.0008 | ±0.0010 | ±0.0015 | ±0.0020 | ±0.0022 |
| | 3 | ±0.0001 | ±0.0005 | ±0.0010 | ±0.0006 | ±0.0007 | ±0.0017 | ±0.0041 | ±0.0030 |

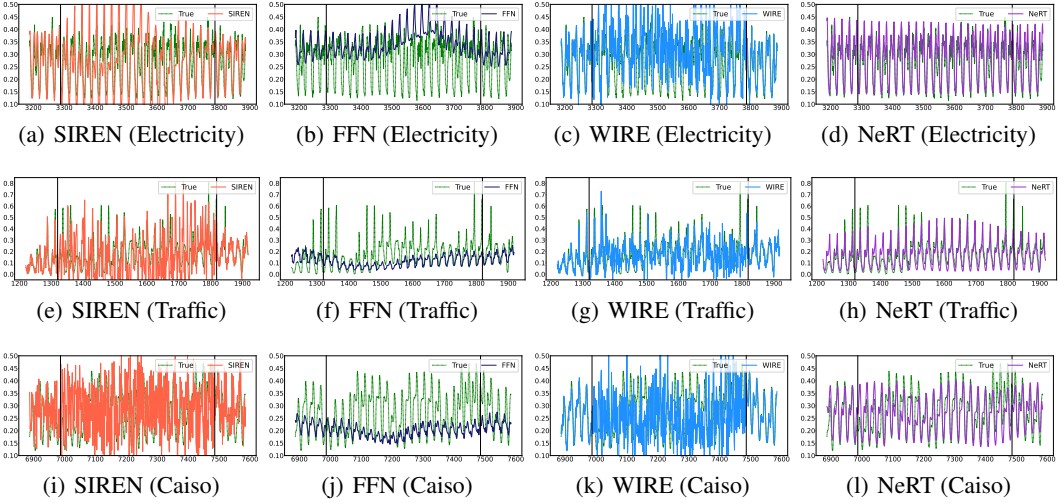

Figure 18: **Interpolation results on missing intervals at the lowest validation error checkpoint.** Interpolation results in Electricity (Figures 18 (a)-(d)), in Traffic (Figures 18 (e)-(h)), and in Caiso (Figures 18 (i)-(l)). The space between the two solid lines represents the testing range, while the outer two parts represent the training range.

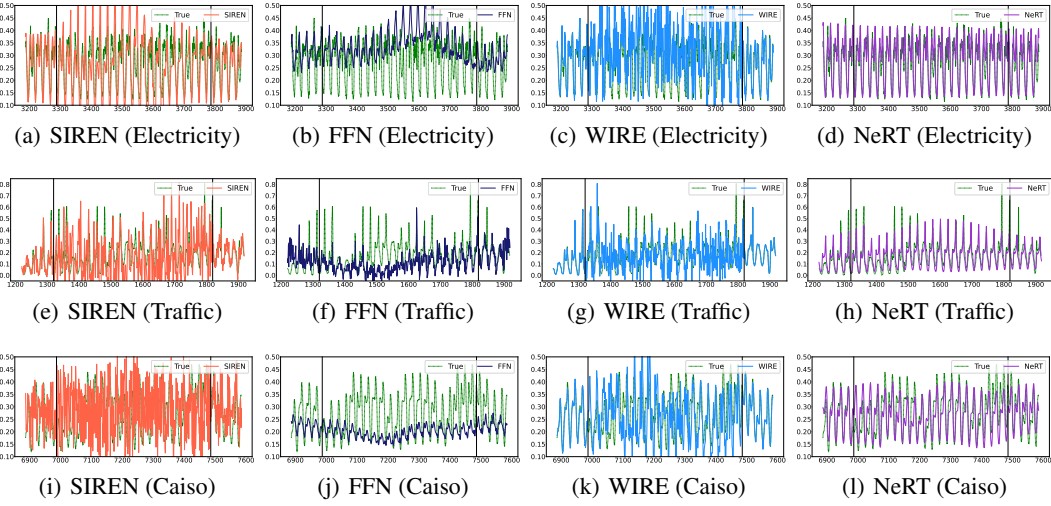

Figure 19: **Interpolation results on missing intervals after the last epoch.** Interpolation results in Electricity (Figures 19 (a)-(d)), in Traffic (Figures 19 (e)-(h)), and in Caiso (Figures 19 (i)-(l)). The space between the two solid lines represents the testing range, while the outer two parts represent the training range.

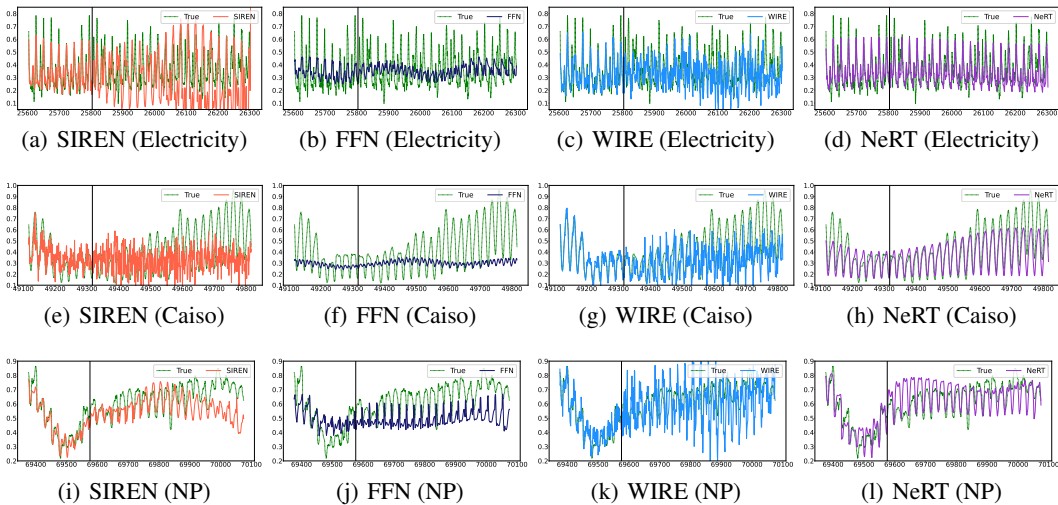

Figure 20: **Extrapolation results on missing intervals at the lowest validation error checkpoint.** Extrapolation results in Electricity (Figures 20 (a)-(d)), in Caiso (Figures 20 (e)-(h)), and in NP (Figures 20 (i)-(l)). The right area after the solid vertical line is a testing range.

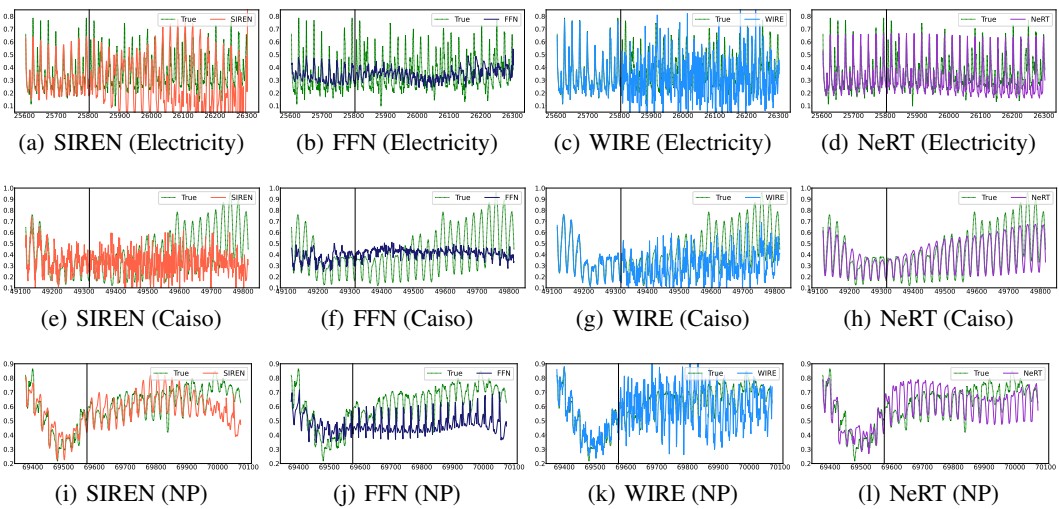

Figure 21: **Extrapolation results on missing intervals after the last epoch.** Extrapolation results in Electricity (Figures 21 (a)-(d)), in Caiso (Figures 21 (e)-(h)), and in NP (Figures 21 (i)-(l)). The right area after the solid vertical line is a testing range.

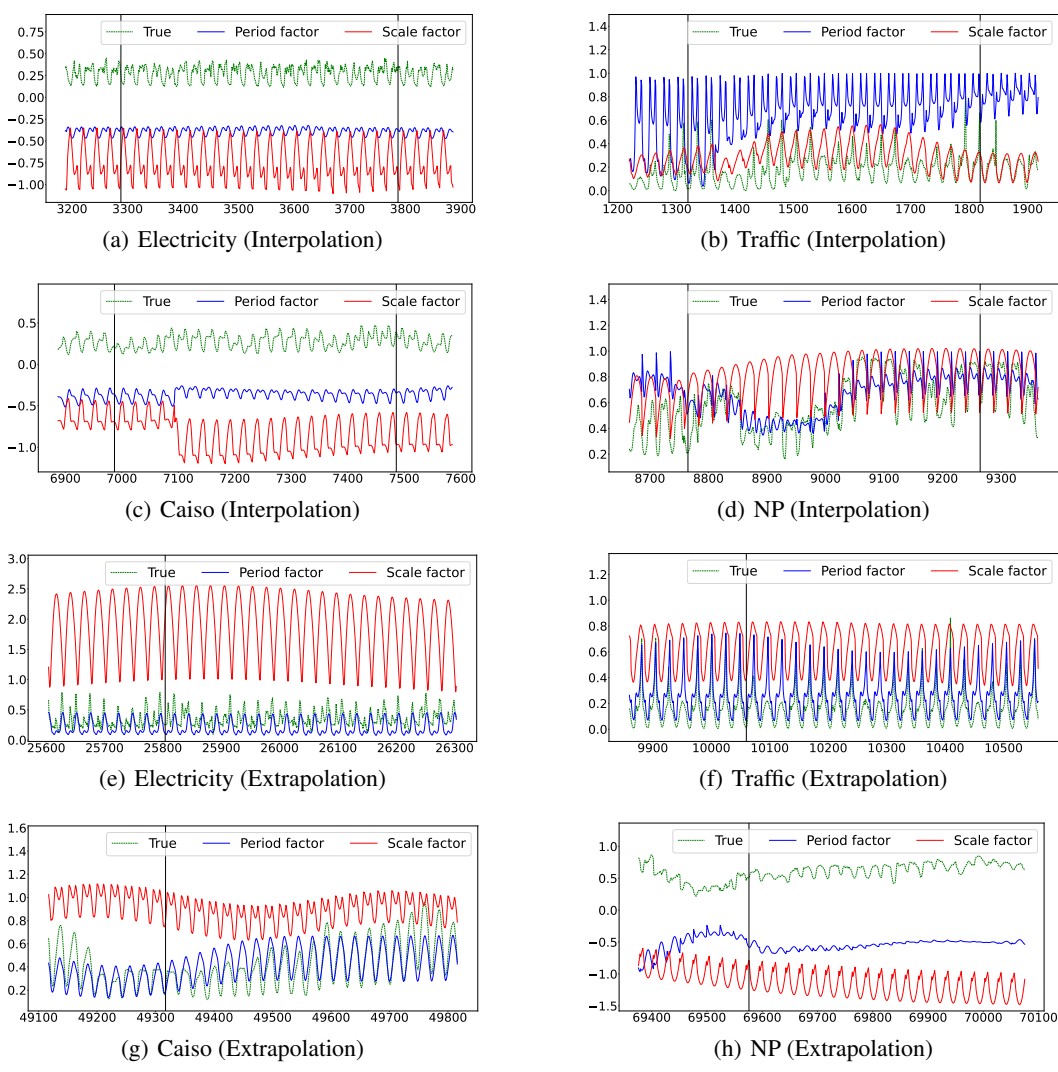

Figure 22: **Periodic and Scale factors trained on periodic time series.** Interpolation results (Figures 22 (a)-(d)), where the space between the two solid lines represents the testing range, while the outer two parts represent the training range, and extrapolation results (Figures 22 (e)-(h)), where right area after the solid vertical line is a testing range.

## J  Additional Comparison with Time Series Baselines

### J.1  Experimental Setups

In this section, we conduct comprehensive analyses between our model and existing time series baselines to support the necessity of adapting INR specifically to time series data (cf. Appendix B). We use four datasets used in Section 4.2, and we have meticulously arranged the experimental setup in the subsequent manner for the purpose of this study. To be specific, we compare the forecasting performance by varying the input window size $m$ of $\{48, 96, 192\}$ and the output window size $n$ of $\{96, 192, 336, 720\}$, following the overall experimental configuration of (Zeng et al., 2022). For each data sample, we fix the total length to 2,880 with a train size of 1,440 and a validation and test sizes of 720. To assess the model performance, we use mean-squared error (MSE) of the test range at the epoch where the best MSE on the validation range is achieved.

**Baselines**  To evaluate the performance of NeRT, we compare it with eight existing time series baselines. As representatives of traditional time series models, we use RNN and LSTM as baselines. Additionally, we compare NeRT against models in the long-term time series forecasting domain, including Linear-based models, i.e., Linear, DLinear, and NLinear from Zeng et al. (2022), and Transformer-based baselines, i.e., Autoformer (Wu et al., 2021), Informer (Zhou et al., 2021), and FEDformer (Zhou et al., 2022). Furthermore, we employ Latent ODE (Rubanova et al., 2019) and Neural CDE (Kidger et al., 2020) as Neural ODE-based models.

**Training Methodological Differences between NeRT and Other Baselines**  Existing time series baselines are highly sensitive to varying input and output window sizes (cf. **L2** of Appendix B), so they need repeated experiments for each window size change. In contrast, since our NeRT operates independently of window size, we need to train NeRT only once for each dataset. During the single training, NeRT makes a single set of predictions for all 720 points, and then calculate MSEs for all the output window sizes of 96, 192, 336, and 720, respectively.

### J.2  Experimental Results

We summarize experimental results of time series forecasting depending on the input/output window sizes in Tables 12, 13, and 14. Note that while other time series baselines are trained individually for each combination of the input/output window sizes, NeRT employs only a single model for each dataset. As shown in the tables, NeRT consistently shows the best MSE regardless of the dataset and window size. On top of that, the results of the baselines are highly affected by the input/output window sizes (cf. Appendix B.1), making hard to choose an appropriate window size setting. Figure 23 depicts how MSE changes as $n$ varies, with $m$ fixed to 96 and 192. In every case, NeRT shows the lowest MSE with the lowest slope, compared to other baselines. Additionally, Figure 24 shows how the models predict $n$ values given the input window size $m$ on the Traffic dataset, where $n = 96$ and $m = 48$ in this setting.

**Computational Cost**  In Table 15, we describe models' computational complexity in terms of time and memory (cf. **L4** of Appendix B). We report the complexity for all window combinations of the baselines individually, whereas with the NeRT's capacity to provide results for all combinations at once using a single model, we report the training complexity of the single NeRT model. The results in Table 15 correspond to an input window size of 96, and each value represents the complexity required during the training of a single data sample. For NeRT, we record the average time/memory complexity needed to train a single data sample, with the values in parentheses indicating the total complexity required for training a single model.

As shown in Table 15, the total memory complexity of NeRT is notably smaller than the memory complexity of all RNN/Transformer/Neural ODE-based models when training just one data sample. This reduction in complexity ranges from being 5.5 times smaller to as much as 1,730 times smaller and in certain instances, it is even smaller than those of Linear-based models. For the time complexity, NeRT outperforms Transformer/Neural ODE-based models, but it is slower than Linear/RNN-based models. In summary, considering the forecasting results from Tables 12, 13, and 14, NeRT demonstrates significant forecasting performance improvements, approximately 2 to 5 times better

than existing time series models, while still maintaining a sufficiently fast training time complexity and memory complexity similar to Linear-based models.

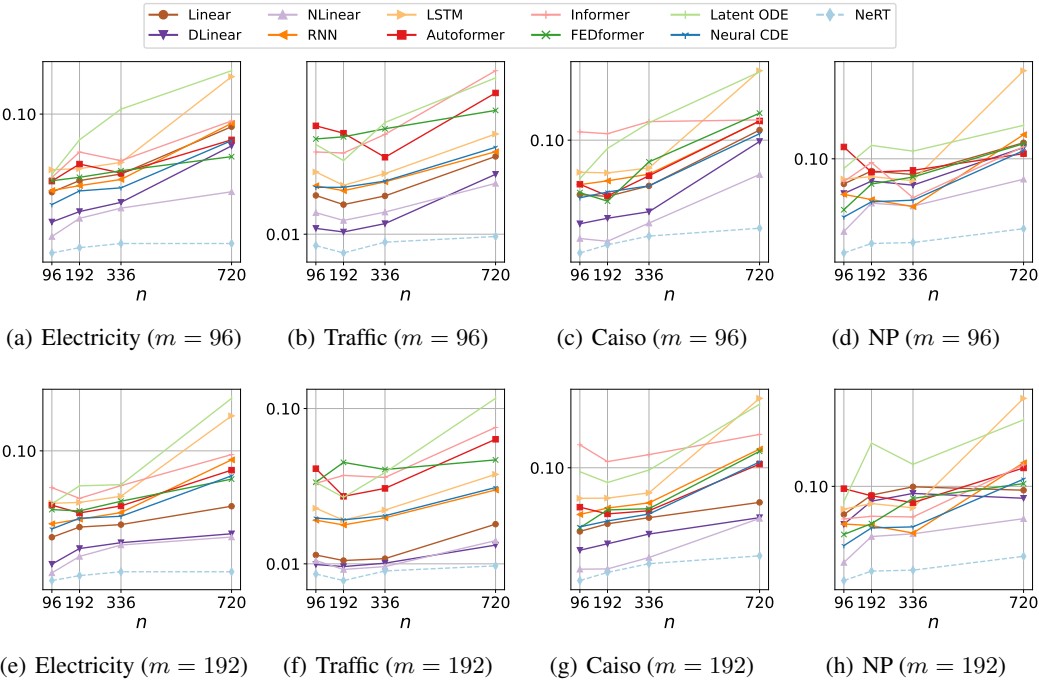

(a) Electricity ($m = 96$)    (b) Traffic ($m = 96$)    (c) Caiso ($m = 96$)    (d) NP ($m = 96$)

(e) Electricity ($m = 192$)    (f) Traffic ($m = 192$)    (g) Caiso ($m = 192$)    (h) NP ($m = 192$)

Figure 23: Comparisons with time series baselines for varying $n = \{96, 192, 336, 720\}$ (output/prediction window size) with a fixed input window size $m = 96$ ((a)-(d)) and $m = 192$ ((e)-(h)).

Table 12: Comparision with time series baselines ($m = 48$)

| Dataset | $n$ | Linear-based | | | RNN-based | | Transformer-based | | | Neural ODE-based | | INR-based |
|---|---|---|---|---|---|---|---|---|---|---|---|---|
| | | Linear | DLinear | NLinear | RNN | LSTM | Autoformer | Informer | FEDformer | Latent ODE | Neural CDE | NeRT (Ours) |
| Electricity | 96 | 0.0671 | 0.0556 | 0.0329 | 0.0387 | 0.0495 | 0.0555 | 0.0485 | 0.0387 | 0.0673 | 0.0333 | **0.0174** |
| | 192 | 0.0747 | 0.0629 | 0.0351 | 0.0411 | 0.0514 | 0.0544 | 0.0610 | 0.0392 | 0.0856 | 0.0390 | **0.0186** |
| | 336 | 0.0843 | 0.0689 | 0.0387 | 0.0444 | 0.0547 | 0.0520 | 0.0641 | 0.0460 | 0.1500 | 0.0403 | **0.0196** |
| | 720 | 0.1461 | 0.1326 | 0.0482 | 0.0890 | 0.1601 | 0.0741 | 0.1003 | 0.0668 | 0.1565 | 0.0720 | **0.0196** |
| Traffic | 96 | 0.0278 | 0.0240 | 0.0215 | 0.0191 | 0.0229 | 0.0395 | 0.0412 | 0.0290 | 0.0349 | 0.0192 | **0.0086** |
| | 192 | 0.0275 | 0.0225 | 0.0202 | 0.0179 | 0.0192 | 0.0368 | 0.0362 | 0.0329 | 0.0528 | 0.0194 | **0.0078** |
| | 336 | 0.0303 | 0.0249 | 0.0219 | 0.0201 | 0.0224 | 0.0325 | 0.0410 | 0.0315 | 0.0460 | 0.0208 | **0.0090** |
| | 720 | 0.0446 | 0.0418 | 0.0289 | 0.0297 | 0.0377 | 0.0652 | 0.0670 | 0.0439 | 0.1036 | 0.0320 | **0.0097** |
| Caiso | 96 | 0.0989 | 0.0843 | 0.0448 | 0.0556 | 0.0671 | 0.0551 | 0.0917 | 0.0455 | 0.1088 | 0.0501 | **0.0236** |
| | 192 | 0.0972 | 0.0799 | 0.0450 | 0.0607 | 0.0647 | 0.0539 | 0.1322 | 0.0410 | 0.1538 | 0.0533 | **0.0262** |
| | 336 | 0.1154 | 0.0927 | 0.0548 | 0.0658 | 0.0711 | 0.0683 | 0.0824 | 0.0602 | 0.1841 | 0.0573 | **0.0293** |
| | 720 | 0.2106 | 0.1948 | 0.0808 | 0.1276 | 0.2431 | 0.1471 | 0.1876 | 0.1467 | 0.3257 | 0.1103 | **0.0324** |
| NP | 96 | 0.1114 | 0.1085 | 0.0690 | 0.0697 | 0.0790 | 0.0784 | 0.0703 | 0.0598 | 0.1060 | 0.0562 | **0.0370** |
| | 192 | 0.1007 | 0.0980 | 0.0795 | 0.0649 | 0.0814 | 0.0733 | 0.0803 | 0.0747 | 0.1522 | 0.0562 | **0.0409** |
| | 336 | 0.1072 | 0.1035 | 0.0932 | 0.0606 | 0.0794 | 0.0860 | 0.0727 | 0.0849 | 0.1465 | 0.0669 | **0.0413** |
| | 720 | 0.2223 | 0.2217 | 0.1414 | 0.1292 | 0.2534 | 0.1073 | 0.0984 | 0.0820 | 0.3414 | 0.1120 | **0.0478** |

Table 13: Comparision with time series baselines ($m = 96$)

| Dataset | $n$ | Linear-based | | | RNN-based | | Transformer-based | | | Neural ODE-based | | INR-based |
|---|---|---|---|---|---|---|---|---|---|---|---|---|
| | | Linear | DLinear | NLinear | RNN | LSTM | Autoformer | Informer | FEDformer | Latent ODE | Neural CDE | NeRT (Ours) |
| Electricity | 96 | 0.0374 | 0.0257 | 0.0214 | 0.0382 | 0.0496 | 0.0430 | 0.0450 | 0.0432 | 0.0468 | 0.0320 | **0.0174** |
| | 192 | 0.0433 | 0.0293 | 0.0269 | 0.0406 | 0.0503 | 0.0533 | 0.0620 | 0.0451 | 0.0723 | 0.0380 | **0.0186** |
| | 336 | 0.0470 | 0.0329 | 0.0306 | 0.0439 | 0.0544 | 0.0476 | 0.0556 | 0.0490 | 0.1064 | 0.0395 | **0.0196** |
| | 720 | 0.0853 | 0.0670 | 0.0376 | 0.0890 | 0.1601 | 0.0723 | 0.0917 | 0.0585 | 0.1726 | 0.0716 | **0.0196** |
| Traffic | 96 | 0.0167 | 0.0108 | 0.0133 | 0.0190 | 0.0228 | 0.0420 | 0.0297 | 0.0353 | 0.0328 | 0.0186 | **0.0086** |
| | 192 | 0.0148 | 0.0103 | 0.0120 | 0.0178 | 0.0191 | 0.0381 | 0.0293 | 0.0363 | 0.0265 | 0.0186 | **0.0078** |
| | 336 | 0.0166 | 0.0115 | 0.0134 | 0.0200 | 0.0223 | 0.0277 | 0.0375 | 0.0404 | 0.0439 | 0.0202 | **0.0090** |
| | 720 | 0.0280 | 0.0221 | 0.0196 | 0.0298 | 0.0377 | 0.0649 | 0.0871 | 0.0515 | 0.0792 | 0.0316 | **0.0097** |
| Caiso | 96 | 0.0498 | 0.0343 | 0.0284 | 0.0566 | 0.0663 | 0.0569 | 0.1112 | 0.0511 | 0.0611 | 0.0478 | **0.0236** |
| | 192 | 0.0487 | 0.0368 | 0.0274 | 0.0594 | 0.0658 | 0.0490 | 0.1086 | 0.0458 | 0.0900 | 0.0515 | **0.0262** |
| | 336 | 0.0557 | 0.0400 | 0.0346 | 0.0647 | 0.0700 | 0.0634 | 0.1268 | 0.0757 | 0.1249 | 0.0557 | **0.0293** |
| | 720 | 0.1138 | 0.0983 | 0.0644 | 0.1273 | 0.2432 | 0.1279 | 0.1296 | 0.1413 | 0.2397 | 0.1086 | **0.0324** |
| NP | 96 | 0.0767 | 0.0693 | 0.0464 | 0.0687 | 0.0807 | 0.1133 | 0.0787 | 0.0584 | 0.0903 | 0.0541 | **0.0370** |
| | 192 | 0.0879 | 0.0790 | 0.0626 | 0.0650 | 0.0828 | 0.0868 | 0.0961 | 0.0766 | 0.1152 | 0.0635 | **0.0409** |
| | 336 | 0.0847 | 0.0757 | 0.0607 | 0.0604 | 0.0794 | 0.0885 | 0.0663 | 0.0825 | 0.1084 | 0.0646 | **0.0413** |
| | 720 | 0.1185 | 0.1126 | 0.0806 | 0.1290 | 0.2534 | 0.1056 | 0.1136 | 0.1173 | 0.1424 | 0.1090 | **0.0478** |

Table 14: Comparision with time series baselines ($m = 192$)

| Dataset | $n$ | Linear-based | | | RNN-based | | Transformer-based | | | Neural ODE-based | | INR-based |
|---|---|---|---|---|---|---|---|---|---|---|---|---|
| | | Linear | DLinear | NLinear | RNN | LSTM | Autoformer | Informer | FEDformer | Latent ODE | Neural CDE | NeRT (Ours) |
| Electricity | 96 | 0.0312 | 0.0217 | 0.0193 | 0.0374 | 0.0493 | 0.0481 | 0.0609 | 0.0453 | 0.0489 | 0.0349 | **0.0174** |
| | 192 | 0.0358 | 0.0268 | 0.0241 | 0.0398 | 0.0499 | 0.0433 | 0.0527 | 0.0445 | 0.0624 | 0.0403 | **0.0186** |
| | 336 | 0.0370 | 0.0290 | 0.0281 | 0.0435 | 0.0542 | 0.0476 | 0.0626 | 0.0507 | 0.0634 | 0.0414 | **0.0196** |
| | 720 | 0.0474 | 0.0327 | 0.0313 | 0.0886 | 0.1602 | 0.0771 | 0.0951 | 0.0683 | 0.2029 | 0.0714 | **0.0196** |
| Traffic | 96 | 0.0114 | 0.0099 | 0.0105 | 0.0191 | 0.0228 | 0.0410 | 0.0330 | 0.0335 | 0.0335 | 0.0197 | **0.0086** |
| | 192 | 0.0105 | 0.0096 | 0.0092 | 0.0178 | 0.0191 | 0.0272 | 0.0371 | 0.0450 | 0.0268 | 0.0192 | **0.0078** |
| | 336 | 0.0108 | 0.0101 | 0.0096 | 0.0198 | 0.0222 | 0.0306 | 0.0360 | 0.0405 | 0.0387 | 0.0204 | **0.0090** |
| | 720 | 0.0180 | 0.0132 | 0.0141 | 0.0299 | 0.0377 | 0.0634 | 0.0756 | 0.0467 | 0.1163 | 0.0310 | **0.0097** |
| Caiso | 96 | 0.0443 | 0.0348 | 0.0273 | 0.0550 | 0.0676 | 0.0604 | 0.1342 | 0.0464 | 0.0951 | 0.0469 | **0.0236** |
| | 192 | 0.0488 | 0.0378 | 0.0274 | 0.0597 | 0.0678 | 0.0554 | 0.1082 | 0.0583 | 0.0828 | 0.0506 | **0.0262** |
| | 336 | 0.0528 | 0.0428 | 0.0317 | 0.0638 | 0.0725 | 0.0577 | 0.1183 | 0.0592 | 0.0971 | 0.0553 | **0.0293** |
| | 720 | 0.0642 | 0.0528 | 0.0523 | 0.1270 | 0.2431 | 0.1048 | 0.1533 | 0.1234 | 0.2253 | 0.1077 | **0.0324** |
| NP | 96 | 0.0743 | 0.0670 | 0.0448 | 0.0672 | 0.0786 | 0.0976 | 0.0709 | 0.0602 | 0.0852 | 0.0535 | **0.0370** |
| | 192 | 0.0912 | 0.0855 | 0.0589 | 0.0661 | 0.0834 | 0.0905 | 0.0729 | 0.0675 | 0.1579 | 0.0645 | **0.0409** |
| | 336 | 0.0995 | 0.0928 | 0.0606 | 0.0611 | 0.0797 | 0.0842 | 0.0724 | 0.0881 | 0.1261 | 0.0652 | **0.0413** |
| | 720 | 0.0960 | 0.0882 | 0.0711 | 0.1282 | 0.2530 | 0.1214 | 0.1254 | 0.1029 | 0.2017 | 0.1079 | **0.0478** |

Table 15: Computational cost with $m$ at 96. Each value is measured during training one data sample. For NeRT, since it is not trained for each window combination, we report the average cost required for training one sample, with total amount in the parentheses.

| Complexity | $n$ | Linear-based | | | RNN-based | | Transformer-based | | | Neural ODE-based | | INR-based |
|---|---|---|---|---|---|---|---|---|---|---|---|---|
| | | Linear | DLinear | NLinear | RNN | LSTM | Autoformer | Informer | FEDformer | Latent ODE | Neural CDE | NeRT |
| Time (sec) | 96 | 0.0333 | 0.0449 | 0.0353 | 0.0864 | 0.0951 | 0.9343 | 0.8525 | 9.3507 | 15.0144 | 9.9457 | 0.2345 |
| | 192 | 0.0329 | 0.0446 | 0.0347 | 0.0886 | 0.0929 | 1.0673 | 0.7802 | 9.4309 | 17.2511 | 9.9699 | (2.8151) |
| | 336 | 0.0320 | 0.0443 | 0.0338 | 0.0898 | 0.0930 | 1.2836 | 0.9100 | 10.7392 | 20.3422 | 9.8612 | |
| | 720 | 0.0202 | 0.0275 | 0.0226 | 0.0335 | 0.0387 | 0.9745 | 0.7102 | 7.7706 | 9.6910 | 5.7095 | |
| Memory (MB) | 96 | 1.0757 | 1.1816 | 1.0991 | 54.2866 | 92.9487 | 888.9063 | 504.9854 | 2254.0459 | 24.2246 | 9.6597 | 0.1318 |
| | 192 | 1.1519 | 1.3169 | 1.1753 | 54.3799 | 93.0425 | 1292.9297 | 679.2822 | 2252.9355 | 26.4951 | 9.6597 | (1.5820) |
| | 336 | 1.2769 | 1.5303 | 1.3003 | 54.5508 | 93.2148 | 1768.7114 | 908.2749 | 2890.7788 | 29.9116 | 9.9468 | |
| | 720 | 1.4126 | 1.8643 | 1.4292 | 46.4707 | 78.8481 | 2735.4766 | 1273.8496 | 3913.7681 | 33.2192 | 8.7642 | |

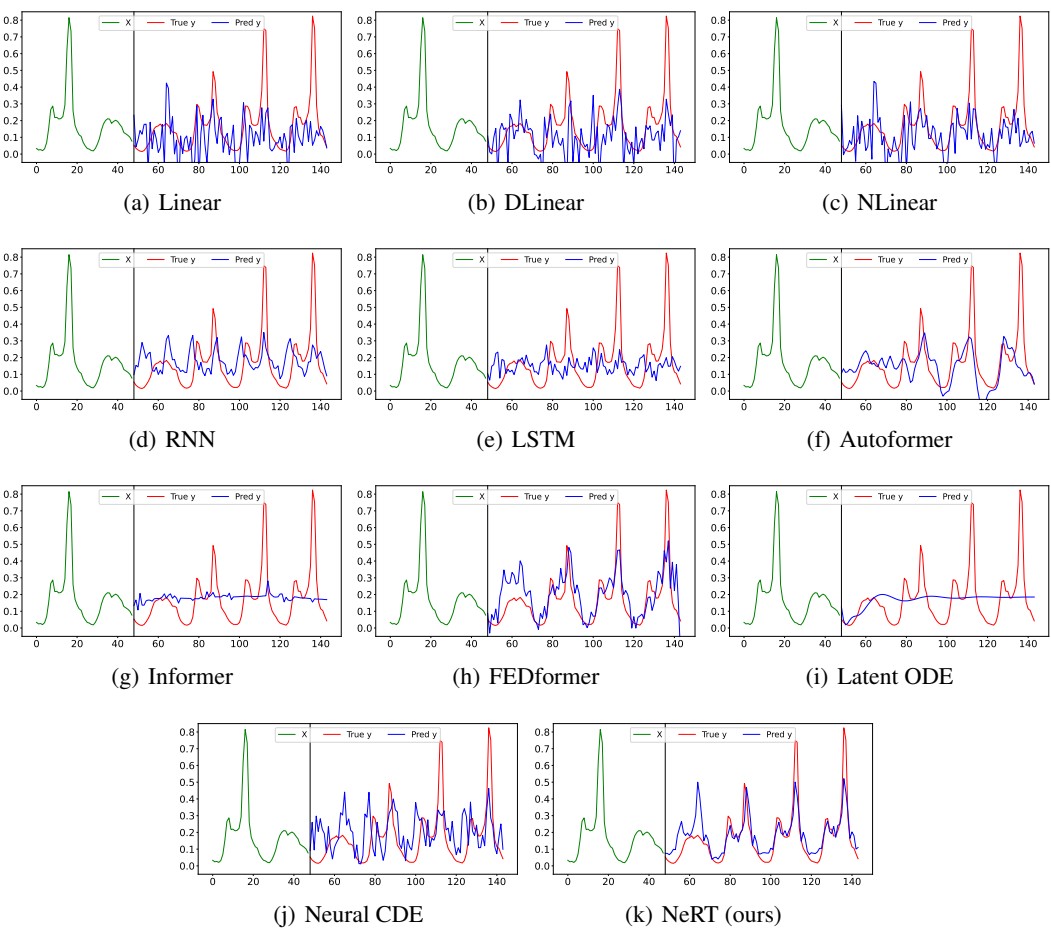

Figure 24: **Forecasting task on Traffic.** We set $m$ to 48 and $n$ to 96. The left side of the solid line represents the input window, while the right side represents the output window.

# K  ADDITIONAL COMPARISON WITH MODULATED INR ON UNSEEN SAMPLES

Table 16: Comparison with Modulated INRs

| Dataset | Task | Modulated SIREN | Modulated FFN | Modulated NeRT (scale) | Modulated NeRT (scale and period) |
|---------|------|-----------------|---------------|------------------------|-----------------------------------|
| Electricity | Imputation | 0.1840 | 0.0355 | 0.0322 | **0.0315** |
| | Forecasting | 0.1844 | 0.0188 | 0.0161 | **0.0149** |
| Traffic | Imputation | 0.0457 | 0.0067 | **0.0009** | 0.0040 |
| | Forecasting | 0.0373 | 0.0116 | **0.0042** | 0.0091 |
| Caiso | Imputation | 0.0306 | 0.0390 | 0.0375 | **0.0264** |
| | Forecasting | 0.0869 | 0.0907 | 0.0663 | **0.0629** |
| NP | Imputation | 0.0361 | 0.0257 | **0.0224** | 0.0229 |
| | Forecasting | 0.0238 | 0.0449 | 0.0441 | **0.0166** |

## K.1  LATENT MODULATION

Fundamentally, a single INR model tends to overfit to a single data sample, making it challenging to represent unseen data samples effectively. To overcome this limitation, recently developed modulation techniques involve i) sharing model parameters and ii) learning sample-specific parameters, enabling INR models capable of representing various data samples. In particular, the latent modulation, introduced in Dupont et al. (2022), is one of the most effective training methods for INRs to infer unseen samples after learning multiple samples — we strictly follow this training method in this subsection. It is a meta-learning-based modulation approach that allows the representation of diverse data samples by adding an additional learnable bias to each shared MLP layer and for each sample, the biases in all the MLP layers are changed — at the end, all data samples can be somehow learned by the combination of the shared MLP parameters and the sample-specific additional biases. By adopting this concept to all INR-based models used in the paper, which are SIREN, FFN, and NeRT, they are able to predict values of unseen samples.

## K.2  EXPERIMENTAL SETUPS

To ensure a fair comparison, NeRT, SIREN, and FFN all employ the same latent modulation approach. The baselines, referred to as modulated SIREN and modulated FFN, are SIREN and FFN models with latent modulation applied to all layers except the first and last. In the case of modulated NeRT, we propose two variants. Firstly, we apply latent modulation exclusively to the scale decoder, denoted Modulated NeRT denoted Modulated NeRT (scale). Secondly, latent modulation is applied to both the scale decoder and the periodic decoder of vanilla NeRT, (scale and period).

The datasets used in the experiments are the periodic time series datasets discussed in Section 4.2, and the experiments are conducted in an environment identical to that described in Appendix I. The dimensionality of the modulation vector remains consistent at 256 throughout the training of all models. Additionally, testing is carried out on unseen block of unseen samples that are not part of the training process. Both imputation and forecasting tasks are simultaneously inferred within a single model.

## K.3  EXPERIMENTAL RESULTS

All experimental results are summarized in Table 16, and it can be observed that modulated NeRTs outperform the modulated INR baselines significantly across all benchmark datasets. Particularly, for Traffic dataset, Modulated NeRT (scale) exhibits MSE values that are approximately one-seventh the magnitude of the baseline for interpolation and half the magnitude for extrapolation. Consequently, NeRT shows its scalability to unseen samples with commendable performance.

## L  EXPERIMENTS ON LONG-TERM TIME SERIES

### L.1  DETAILED EXPERIMENTAL SETUPS

Table 17: Hyperparameters of long-term time series

|  | $S_{max}$ | Drop ratio | $\omega^{\text{init}}$ | $\omega^{\text{inner}}$ | $\dim(\psi_t)$ | $\dim(\psi_F)$ | $\dim(\mathbf{h}_p)$ | $\dim(\mathbf{h}_s)$ |
|---|---|---|---|---|---|---|---|---|
| ETTh1 | 100 | 30% | 5.0 | 1.0 | 50 | 30 | 200 | 10 |
|  |  | 50% | 5.0 | 3.0 | 30 | 30 | 100 | 50 |
|  |  | 70% | 10.0 | 3.0 | 30 | 30 | 100 | 30 |
| ETTh2 | 100 | 30% | 5.0 | 1.0 | 10 | 30 | 200 | 10 |
|  |  | 50% | 5.0 | 3.0 | 30 | 30 | 100 | 50 |
|  |  | 70% | 10.0 | 3.0 | 10 | 30 | 50 | 30 |
| National Illness | 1 | 30% | 5.0 | 1.0 | 50 | 10 | 100 | 10 |
|  |  | 50% | 5.0 | 3.0 | 30 | 50 | 30 | 10 |
|  |  | 70% | 10.0 | 3.0 | 10 | 10 | 10 | 10 |

For fair comparison, we share $\omega^{\text{init}}$ and $\omega^{\text{inner}}$ and employ similar model sizes across the tested models. We note hyperparameter configurations used in long-term time series experiments in Table 17. In terms of the number of layers in NeRT, we set $L_t$, $L_f$ and $L_s$ to 2, and $L_p$ to 5.

### L.2  ADDITIONAL EXPERIMENTAL RESULTS

Table 18: **Full table on long-term time series.** The best results are reported in boldface.

|  | Drop ratio | Linear | Cubic | SIREN | FFN | NeRT |
|---|---|---|---|---|---|---|
| ETTh1 | 30% | 0.0892 | 0.1268 | $0.1945{\pm}0.0030$ | $0.2522{\pm}0.0392$ | $\mathbf{0.0828}{\pm}\mathbf{0.0028}$ |
|  | 50% | 0.1178 | 0.1662 | $0.2173{\pm}0.0216$ | $0.3407{\pm}0.0133$ | $\mathbf{0.0911}{\pm}\mathbf{0.0097}$ |
|  | 70% | 0.1978 | 0.2902 | $0.2605{\pm}0.0082$ | $0.4256{\pm}0.0199$ | $\mathbf{0.1257}{\pm}\mathbf{0.0056}$ |
| ETTh2 | 30% | 0.0407 | 0.0655 | $0.1010{\pm}0.0075$ | $0.1863{\pm}0.0113$ | $\mathbf{0.0344}{\pm}\mathbf{0.0020}$ |
|  | 50% | 0.0473 | 0.0847 | $0.0723{\pm}0.0021$ | $0.2351{\pm}0.0138$ | $\mathbf{0.0423}{\pm}\mathbf{0.0022}$ |
|  | 70% | 0.0596 | 0.1250 | $0.0964{\pm}0.0024$ | $0.3178{\pm}0.0460$ | $\mathbf{0.0575}{\pm}\mathbf{0.0001}$ |
| National Illness | 30% | 0.0266 | 0.0248 | $0.3502{\pm}0.0210$ | $0.1110{\pm}0.0059$ | $\mathbf{0.0239}{\pm}\mathbf{0.0109}$ |
|  | 50% | 0.0567 | 0.0484 | $0.1716{\pm}0.0376$ | $0.2319{\pm}0.0737$ | $\mathbf{0.0291}{\pm}\mathbf{0.0048}$ |
|  | 70% | 0.0902 | 0.0876 | $0.3564{\pm}0.0202$ | $0.4453{\pm}0.0442$ | $\mathbf{0.0871}{\pm}\mathbf{0.0257}$ |

We report the full experimental results of Table 2 in Table 18. As shown in Table 18, our NeRT shows the lowest MSE in every dataset, regardless of the drop ratio. For example, NeRT shows an MSE of 0.1257 in ETTh1 with a drop ratio of 70%, while baselines exhibit errors from 0.1978 in minimum to 0.4256 in maximum. Figures 25, 26, and 27 show how models learn and represent the spatiotemporal coordinates of ETTh1, ETTh2, and National Illness, respectively. In those figures, the top row ((a)-(c)) distinguishes the training and the testing sets in the learned coordinate systems where the X-axis refers to the temporal information and the Y-axis is the spatial information. While training, models only see the white-colored coordinates, i.e., training samples, and then predict values in the black-colored coordinates, i.e., validating and testing samples. Other rows are the results of the first 50 timestamps by each method in each dataset. Unlike other baselines, NeRT successfully learns the spatial coordinate systems to embed features and accurately represents the temporal pattern in each feature. Surprisingly, NeRT demonstrates remarkable predictions even in extreme scenarios with a drop ratio of 70%, and it maintains its performance well compared to other baselines in challenging situations, i.e., high drop ratios.

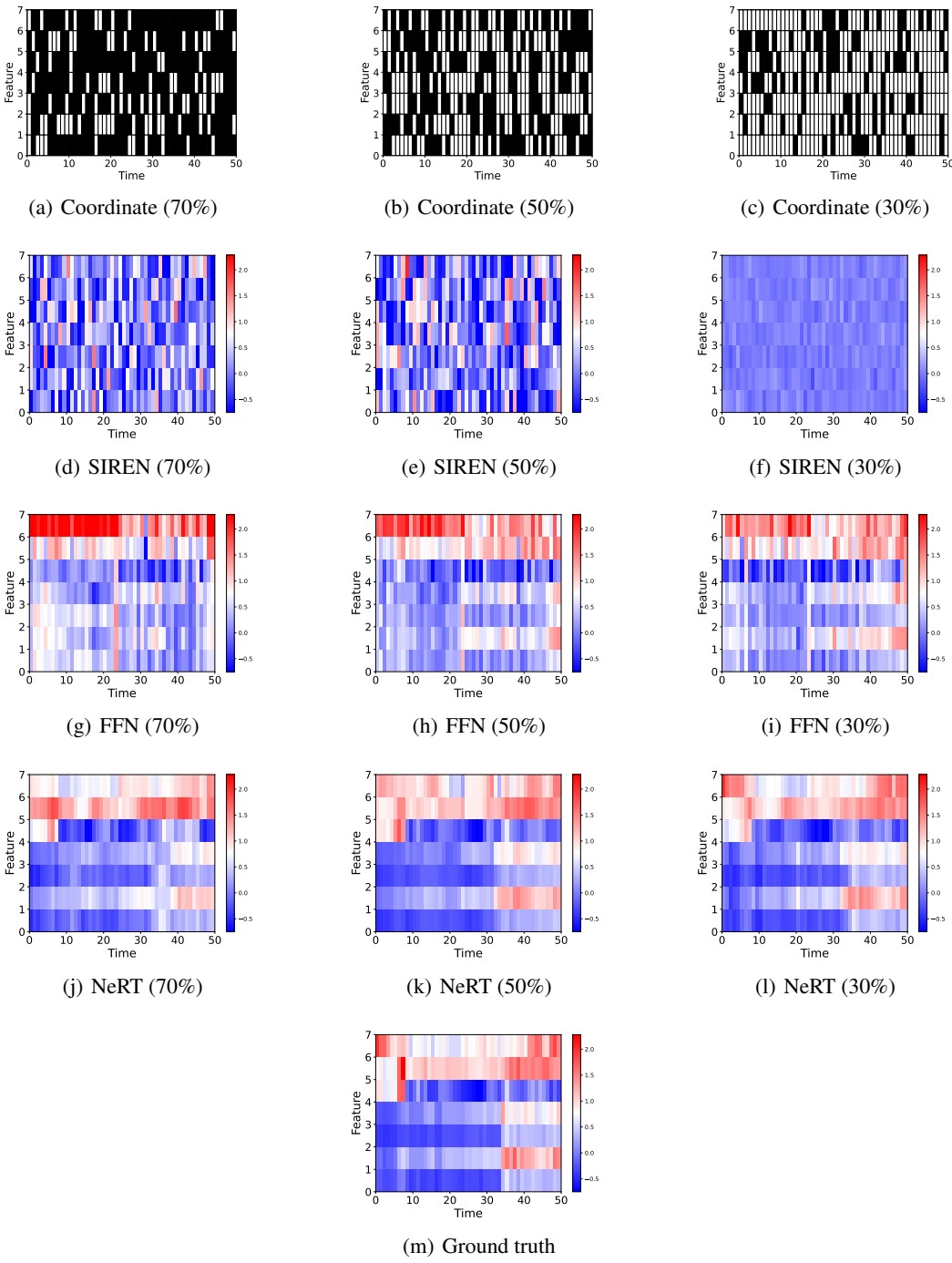

Figure 25: Experimental results of long-term time series (ETTh1). In (a)-(c), white (resp. black) cells mean training (resp. validating/testing) samples.

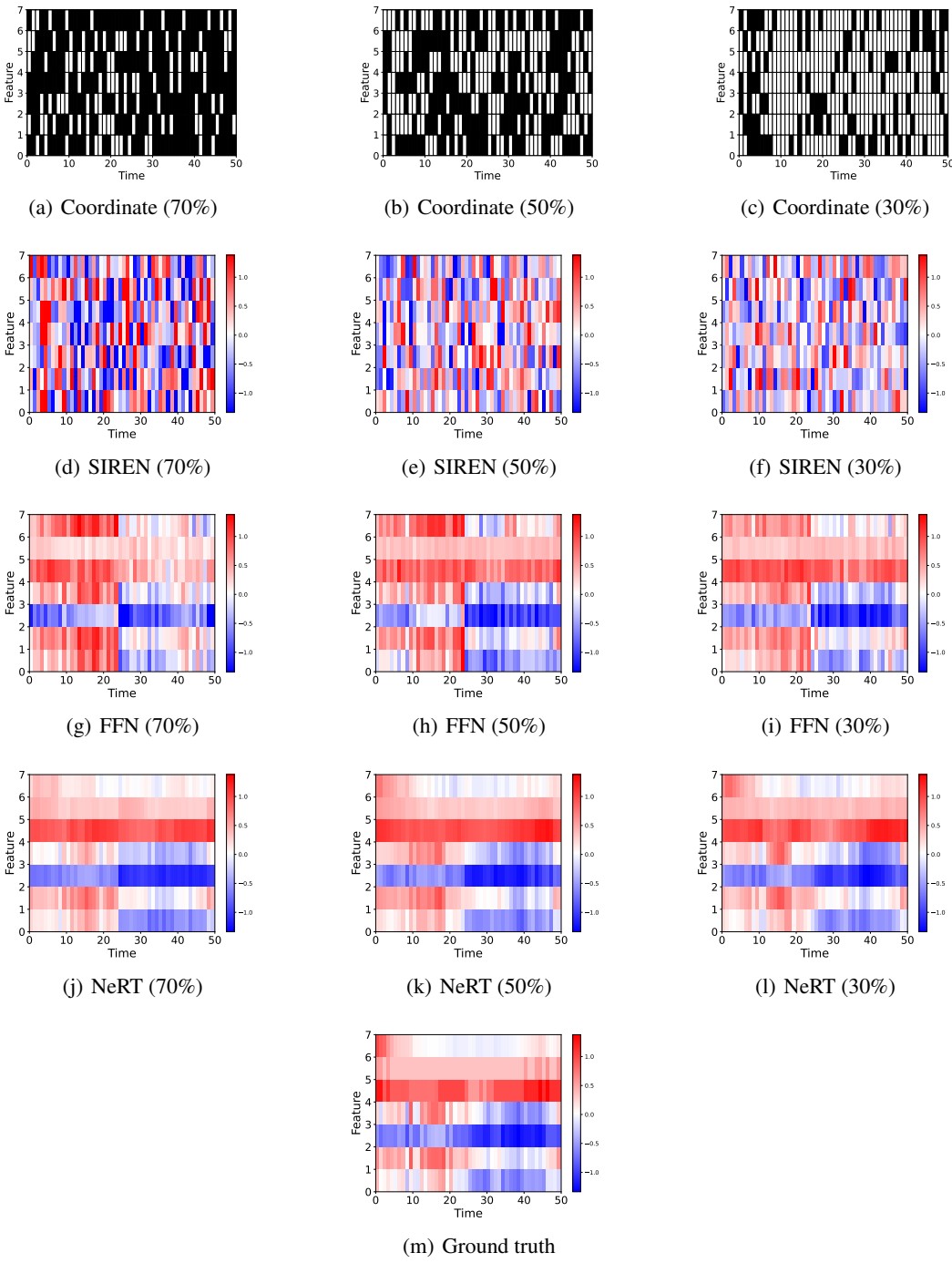

Figure 26: Experimental results of long-term time series (ETTh2). In (a)-(c), white (resp. black) cells mean training (resp. validating/testing) samples.

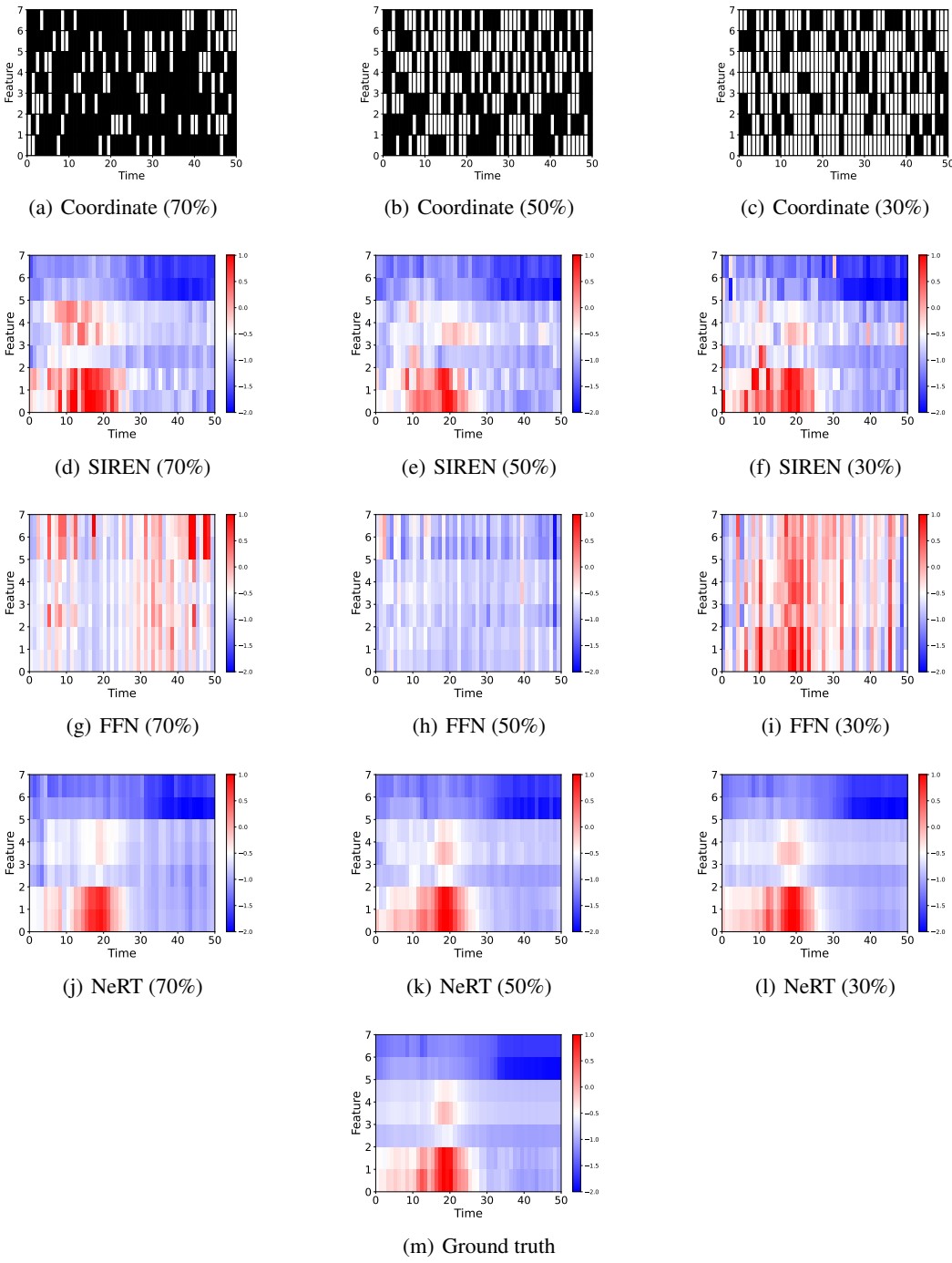

Figure 27: Experimental results of long-term time series (National Illness). In (a)-(c), white (resp. black) cells mean training (resp. validating/testing) samples.

# M    EXPERIMENTS ON COUPLED MASS-SPRING SYSTEM

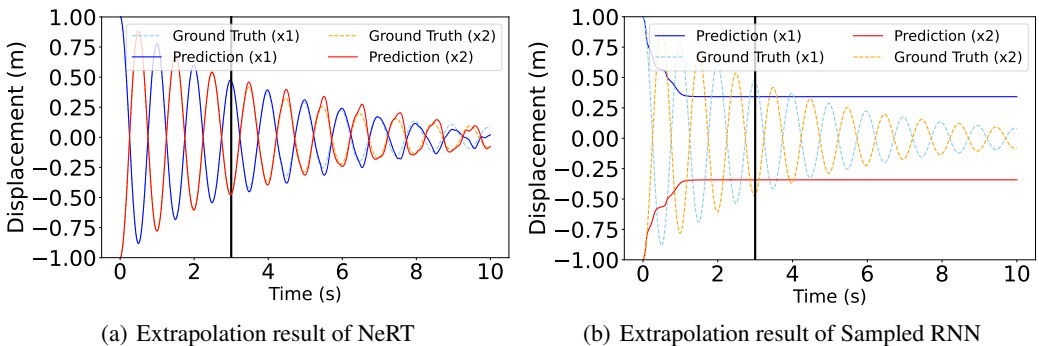

(a) Extrapolation result of NeRT       (b) Extrapolation result of Sampled RNN

Figure 28: **Coupled mass-spring system.** Extrapolation results (Figures 28 (a)-(b)). The left side of the solid vertical line represents the training range, while the right side represents the testing range.

In this section, we present an additional experiment to evaluate NeRT's performance using a coupled mass-spring system. This system demonstrates the dynamics of two masses interacting strongly through springs, providing a classical physics benchmark for studying oscillatory motion. For a simple two mass system, the form of the governing equation is as follows:

$$
\begin{aligned}
m_1\ddot{x}_1 &= -k_1x_1 - k_2(x_1 - x_2) - c_1\dot{x}_1, \\
m_2\ddot{x}_2 &= -k_2(x_2 - x_1) - k_3x_2 - c_2\dot{x}_2,
\end{aligned}
\tag{6}
$$

where $x_1$ and $x_2$ represent the displacements of masses $m_1$ and $m_2$ from their equilibrium positions, $\ddot{x}_1$ and $\ddot{x}_2$ denote their accelerations, and $\dot{x}_1$ and $\dot{x}_2$ represent their velocities. The terms $k_1, k_2$ and $k_3$ are the spring constants, and $c_1$ and $c_2$ are the damping coefficients. Mass $m_1$ is connected to a wall via a spring with constant $k_1$, and to mass $m_2$ via a coupling spring with constant $k_2$. Similarly mass $m_2$ is connected to another wall through a spring with constant $k_3$. The systems is initialized with the following parameters: both masses $m_1, m_2$ are set to $1.0kg$, the spring constants are $k_1 = 10.0N/m$, $k_2 = 15.0N/m$, and $k_3 = 10.0N/m$, and the damping coefficients are $c_1 = 0.5kg/s$ and $c_2 = 0.5kg/s$. The initial displacements are $x_1(0) = 1.0m$ and $x_2(0) = -1.0m$, with initial velocities set to zero.

For this experiment, we train NeRT using data from $t = 0s$ to $t = 3s$ and evaluate its ability to extrapolate the system dynamics from $t = 3s$ to $t = 10s$. As shown in Figure 28, the experimental results demonstrate that NeRT can generalize effectively modeling the dynamics of the coupled system over extended time periods. To validate performance of NeRT, we compared it with Sampled RNN, a state-of-the-art RNN-based method. NeRT achieved a lower MSE during extrapolation (0.0012) compared to Sampled RNN (0.2632), demonstrating stronger generalization beyond the training range.

# N    EXPERIMENTS ON CHAOTIC DYNAMIC: LORENZ SYSTEM

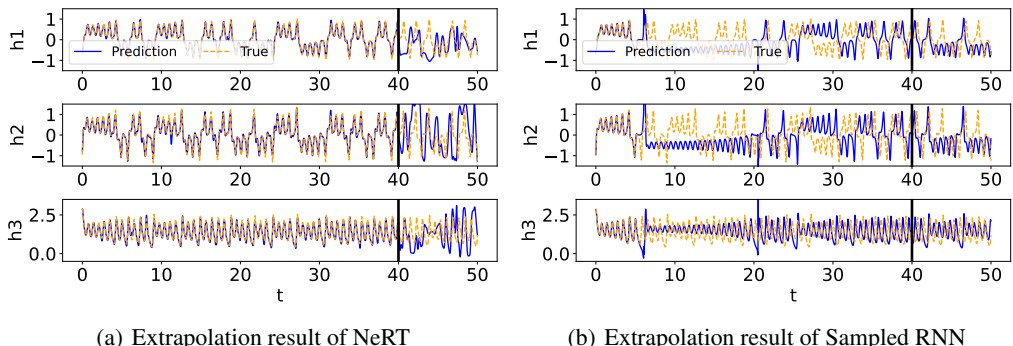

(a) Extrapolation result of NeRT          (b) Extrapolation result of Sampled RNN

Figure 29: **Lorenz system.** Extrapolation result of NeRT (Figure 29 (a)) and extrapolation result of Sampled RNN(Figure 29 (b)). The left side of the solid vertical line represents the training range, while the right side represents the testing range.

To further evaluate the broader applicability of NeRT, we conducted additional experiments on the chaotic Lorenz system. The Lorenz system, originally devised to model atmospheric convection in Lorenz (1963), is governed by the following set of equations:

$$\dot{h_1} = \sigma(h_2 - h_1)$$
$$\dot{h_2} = h_1(\rho - h_3) - h_2 \quad\quad (7)$$
$$\dot{h_3} = h_1 h_2 - \beta h_3,$$

where $\sigma, \rho, \beta$ are parameters that control the dynamics of the system. For our experiments, we set $\sigma = 10$, $\beta = 8/3$, and $\rho = 28$, which corresponds to the chaotic regime of the Lorenz system. Specifically, the dataset is generated by solving the initial value problem for $t \in [0, 5]$ with a time step of $\Delta t = 0.01$. Initial conditions are sampled randomly from a uniform distribution, $h_0 \sim \text{Uniform}([-20, 20] \times [0, 50])$.

For baseline comparison, we include the state-of-the-art RNN-based model, Sampled RNN (Bolager et al., 2024), which has shown strong performance in modeling chaotic dynamic systems. We use the data from $t = 0$ to $t = 40s$ for training, and the data from $t = 40s$ to $t = 50s$ for testing. As shown in Figure 29, within the training region, NeRT successfully models the system's dynamics, demonstrating its ability to capture chaotic behavior over shorter time intervals. However, both NeRT and Sampled RNN, struggle with the extrapolation task for the Lorenz system.

The Lorenz system, as a chaotic system, poses significant challenges in extracting periodicity, which highlights the limitations of both models. This dataset serves as a valuable tool for identifying these limitations and emphasizes the need to extend the underlying framework to handle non-periodic signals. Expanding this methodology to model non-periodic systems remains a critical avenue for future work, offering potential for broader applicability to complex, real-world dynamics.

## O  ADDITIONAL COMPARISON WITH GAUSSIAN PROCESS

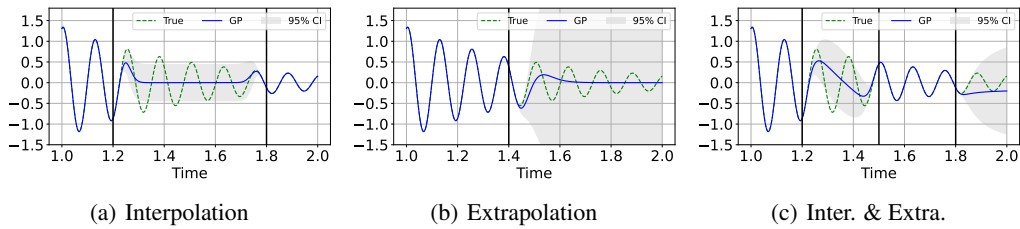

(a) Interpolation  (b) Extrapolation  (c) Inter. & Extra.

Figure 30: **Results of the Gaussian Process (GP) with rational quadratic kernel on the damped oscillation ODE.** Figures (a), (b) and (c) show the results for interpolation, extrapolation, and both interpolation and extrapolation, respectively.

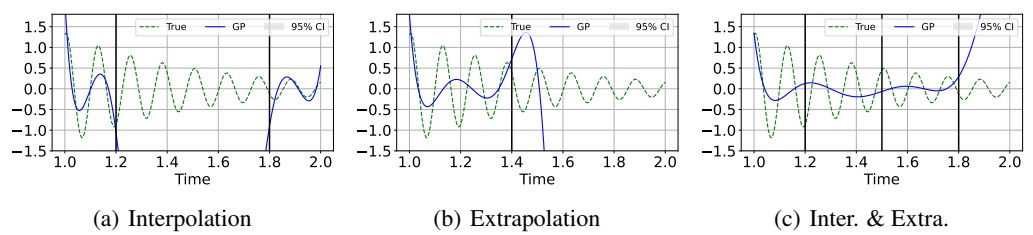

(a) Interpolation  (b) Extrapolation  (c) Inter. & Extra.

Figure 31: **Results of the Gaussian Process (GP) with RBF kernel on the damped oscillation ODE.** Figures (a), (b) and (c) show the results for interpolation, extrapolation, and both interpolation and extrapolation, respectively.

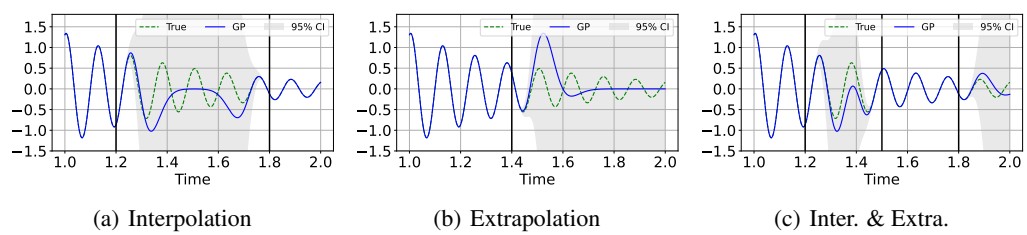

(a) Interpolation  (b) Extrapolation  (c) Inter. & Extra.

Figure 32: **Results of the Gaussian Process (GP) with RBF+White kernel on the damped oscillation ODE.** Figures (a), (b) and (c) show the results for interpolation, extrapolation, and both interpolation and extrapolation, respectively.

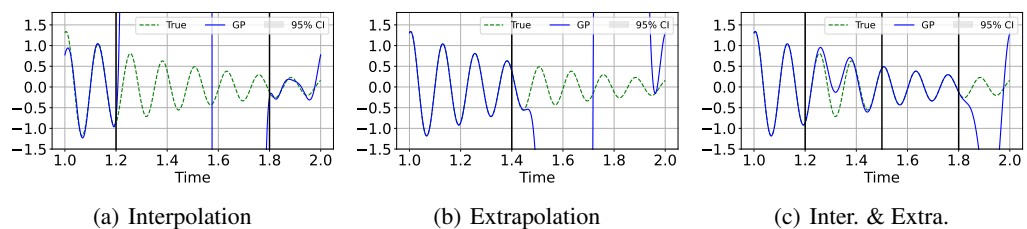

(a) Interpolation  (b) Extrapolation  (c) Inter. & Extra.

Figure 33: **Results of the Gaussian Process (GP) with exponential sine squared kernels on the damped oscillation ODE.** Figures (a), (b) and (c) show the results for interpolation, extrapolation, and both interpolation and extrapolation, respectively.

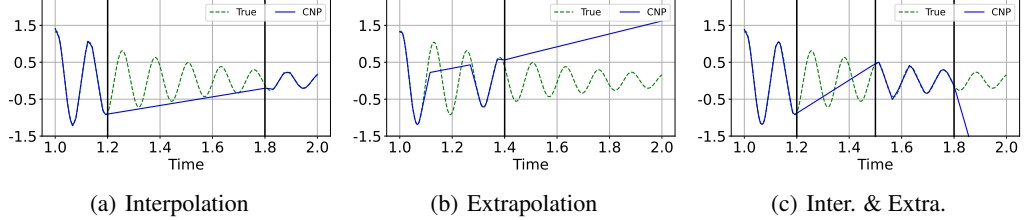

(a) Interpolation        (b) Extrapolation        (c) Inter. & Extra.

Figure 34: **Results of the Conditional Neural Processes (CNP) (Garnelo et al., 2018) on the damped oscillation ODE.** Figures (a), (b) and (c) show the results for interpolation, extrapolation, and both interpolation and extrapolation, respectively.

We conduct additional experiments to compare the performance of Gaussian Process (GP) with different kernels and the Conditional Neural Process (CNP) Garnelo et al. (2018) on the damped oscillation dataset. The GP experiments utilize four types of kernels: i) rational quadratic, ii) RBF, iii) RBF+white, iv) and squared exponential periodic. Figures 30, 31, 32 and 33 illustrate the interpolation and extrapolation results for each kernel. Across all cases, the GPs struggle to accurately capture the dynamics. Additionally, we evaluate the CNP model under the same experimental settings (please see Section 4.1.1 and Appendix E) and results are summarized in Figure 34. Similar to GP, CNP faces significant challenges in performing both interpolation and extrapolation.

## P COMPARISON WITH BASELINE FOR IMPUTATION TASK

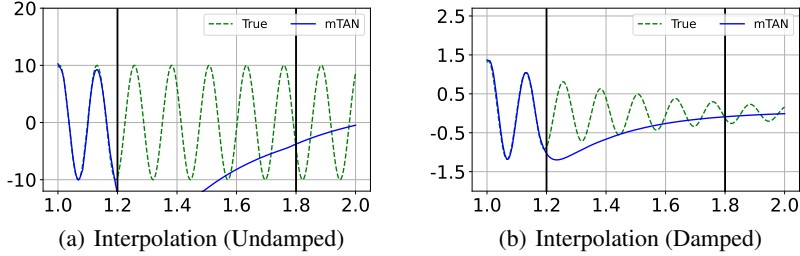

(a) Interpolation (Undamped)        (b) Interpolation (Damped)

Figure 35: **Results of the Multi-time Attention Networks (mTAN) Shukla & Marlin (2021) on the damped/undamped oscillation ODEs.** Extrapolation results (Figures 35 (a)-(b)). The inside of the two solid lines represents the testing range ($t \in [1.2, 1.8]$, while the outside represents the training range ($t \in [1.0, 1.2], [1.8, 2.0]$).

We conduct additional interpolation experiments using the multi-time Attention Networks (mTAN) (Shukla & Marlin, 2021) on undamped and damped oscillation datasets. As shown in Figure 35, mTAN struggles to perform interpolation under these conditions. All experimental settings are consistent with those used for NeRT, ensuring a fair comparison. Detailed experimental settings are provided in Appendix E.

