# OpenReview forum: "Neural Functions for Learning Periodic Signal"
_ICLR.cc/2025/Conference — ICLR 2025 Poster_

### Official Review · Reviewer_CBC1 · 2024-10-28

**Soundness:** 3
**Presentation:** 2
**Contribution:** 2
**Rating:** 8
**Confidence:** 3

**Summary:**

The authors propose to approximate periodic signals using a carefully designed encoder + decoder architecture which can be interpreted as learning a decomposition of the input signal into its frequency and scale components. The authors demonstrate their approach provides improved performance on some INR and time-series forecasting benchmarks.

**Strengths:**

- The proposed architecture could be useful in a number of domains where the underlying data is known to exhibit strong periodic structure.
- The proposed architecture is mathematically sound
- The proposed architecture performs well on the numerical studies chosen by the authors demonstrating that the proposed architecture is a useful addition.

**Weaknesses:**

- While the proposed architecture is a useful contribution, I struggled to understand the specific challenges your approach is trying to address as opposed to classical methods like GP regression with periodic kernels, compressive sensing of periodic signals or even older works for learning periodic signals with neural networks [1]. It would be helpful to be specific about the application domains you are interested in and why classical approaches are not fit for these tasks.

- I think the discussion on how your approach can be used for time-series forecasting could do with some refinement. Typical approaches for time-series forecasting choose to learn both a trend / seasonality component + an autoregressive component because we assume many time-series forecasting problems contain both long term trends + seasonalities and autoregressive dynamics (such as local momentum and mean reversion). I think it would be helpful to be more explicit about the theoretical limitations of using your approach over standard autoregressive models and their variants in this setting.

[1] Ruiz, A., D. H. Owens, and S. Townley. "Learning periodic signals with recurrent neural networks." (1996): 1131-1136.

**Questions:**

- Rather than replacing traditional approaches for time-series forecasting, can you see your approach being used in combination with autoregressive style approaches?

- How does the shift invariance of your approach impact forecasting for time-series problems?

---

### Official Review · Reviewer_DxCf · 2024-11-03

**Soundness:** 3
**Presentation:** 3
**Contribution:** 3
**Rating:** 8
**Confidence:** 3

**Summary:**

This paper introduces a simple but effective strategy to model periodic time-series data. The main contribution of this paper incorporates a Fourier mapping layer to encode the scale and periodic factors and reconstructs the signal via (scale * periodic signals). In the experiments, this work demonstrates robust and good performances on all datasets including synthetic data and real-world atmospheric data.

**Strengths:**

This paper is very well-written with extensive experimental study to demonstrate the proposed method (NeRT). The method is simple to understand and clear to interpret. The Fourier mapping layer can be very helpful to model this kind of periodic behavior from time-series. The experimental section is also very strong, which outperforms all baseline methods. The detailed visualization and ablation study further show how NeRT is a very effective strategy for complex signals with periodic behaviors.

**Weaknesses:**

The weaknesses of this paper are also clear which is mainly coming from the fact that NeRT could only potentially able to model periodic signals. The Fourier embedding/feature may be a huge constraint since the signal must have a fixed period. Yet this is understandable since there are many natural phenomena that has fixed frequency (e.g. the global temperature, traffic).

There are also some issues in terms of presentation that may potentially need to be fixed.

**Questions:**

Even though the paper is very well-written, the reviewer would still like to ask the following clarifying questions during the rebuttal phase. The reviewer is very likely to change the final overall score.

1. Could the authors define what NeRT stands for? The reviewer tried to find a clear definition but was unable to.

2. The authors cited Fan et al. (2020) but did not include their method as a baseline. What might be the reason for this?

3. Could the authors fix the order/placement of the figures? It can be confusing to match the figures with their corresponding sections.

4. The reviewer noticed that the authors mainly conducted experiments on 1D time-series data. Are there any limitations preventing NeRT from being applied to 2D time-series data (e.g., coupled systems)?

5. Can NeRT be generalized to signals with mixed frequencies? In this case, the signals may combine low-, mid-, and high-frequency components.

6. Although methods like RNN and LSTM may require a window, they could be more effective for data with shifting periods (where spikes do not occur at fixed times). How might NeRT adapt to such data? The reviewer is concerned that NeRT may be overly biased toward signals with exact periods, which could be a significant limitation of this approach.

7. (minor suggestion) It might be helpful to further elaborate on Sec. 3.1 when defining the c_i^t and c_i^f. A better wording here could help the audiences to understand the framework even better.

---

### Official Review · Reviewer_NJRt · 2024-11-07

**Soundness:** 3
**Presentation:** 3
**Contribution:** 3
**Rating:** 6
**Confidence:** 4

**Summary:**

The paper presents an approach to improving the performance of coordinate-based multi-layer perceptrons (MLPs), also known as implicit neural representations (INRs), particularly in scenarios where the underlying signals exhibit periodic properties. The authors identify limitations in existing INR architectures, such as overfitting and poor generalization beyond the training region, which lead to suboptimal extrapolation performance, especially for sequential data like time series. To address these challenges, the paper proposes a new network architecture that enhances the ability of INRs to capture periodic patterns. The proposed approach involves two separate internal networks to learn the periodic and scale components of an observed signal.  The proposed algorithm requires only a single trained model to perform both interpolation (imputation) and extrapolation (forecasting). The proposed method demonstrates superior performance over state-of-the-art (SoTA) baselines in interpolation and extrapolation tasks across several synthetic and real-world benchmark datasets.

**Strengths:**

- The paper introduces a novel architecture for implicit neural representation, consisting of two separate internal networks to learn the periodic and scale components of an observed signal.
- The proposed approach enables forecasting and imputation using a single trained model.
- Experiments show that the framework achieves better performance than baselines across multiple tasks and datasets.
- The paper is well written and easy to follow.

**Weaknesses:**

- Could the authors provide insights into why an INR-based approach excels in forecasting and imputation of time series compared to RNN-style approaches that use time series sequences as input? Is the superior performance primarily due to the ability of the proposed method to better capture periodicity present in the datasets?
- The paper currently lacks comparisons with state-of-the-art approaches for the imputation/interpolation task.
- While the paper claims that the proposed approach can model irregularly-sampled time series, there are no experiments on real-world datasets to substantiate this claim. I would like the authors to compare their method with state-of-the-art approaches [1,2] for irregular time series on datasets such as MIMIC-III or PhysioNet to support their argument.

### References
1. Multi-Time Attention Networks for Irregularly Sampled Time Series, International Conference on Learning Representations, 2021.
2. Latent ordinary differential equations for irregularly-sampled time series, Neural Information Processing Systems, 2019.

**Questions:**

Listed in the weaknesses section.

---

### Official Review · Reviewer_Ta2X · 2024-11-09

**Soundness:** 3
**Presentation:** 3
**Contribution:** 2
**Rating:** 6
**Confidence:** 4

**Summary:**

The paper introduces a novel framework called NeRT, designed to model periodic signals effectively. It separates periodic and scale factors in the data, enabling improved predictions for both interpolation and extrapolation tasks using a single model. The proposed method demonstrates its effectiveness through extensive experiments, which include learning periodic solutions for differential equations and performing time series tasks such as imputation (interpolation) and forecasting (extrapolation) with real-world datasets.

**Strengths:**

**Originality**: The NeRT framework models periodic signals by effectively separating periodic and scale factors. This method creatively combines concepts from Fourier analysis and neural network architectures, leading to a more nuanced understanding of periodic behaviors in data.

**Clarity**: The paper is almost well-written and clear. The authors explain the motivation for their approach, the architecture of NeRT, and the experimental results. Diagrams and examples support key points throughout.

**Significance**: The significance of this work lies in its potential to advance the field of time series and sequential data analysis, particularly in applications involving periodic signals/patterns.

**Weaknesses:**

1. Although the proposed method can model/forecast periodic signals, I am not convinced it outperforms state-of-the-art RNNs [2] or other successful methods for analyzing and modeling time series data, such as Koopman-based methods [1]. These methods can model and forecast not only periodic time series but also more challenging time series data related to chaotic systems, see for instance [1,2,3,4,5].
- Could the authors compare more aspects, for example, **training time** and the **hyperparameters** (the number of hyperparameters and computationally consuming hyperparameter search) for their method and the state-of-the-art methods (e.g. shPLRNN in [2]) for simple benchmark systems like the van der Pol oscillator?
Also, please note that while MSE seems an appropriate evaluation measure for periodic time series, for non-periodic time series data, one should consider other measures like Geometrical Measure (D_stsp) and/or Temporal Measure (D_H) as mentioned in [2].

2. While periodic signals are indeed significant in various applications, real-world datasets often exhibit a wide range of behaviors, including non-periodic, chaotic, and irregular patterns. Therefore, designing a method that exclusively targets periodic signals may limit its applicability in practical scenarios where the underlying dynamics of the time series are **unknown** or **complex**.
- Could the authors test NeRT on non-periodic time series data, for example, on chaotic systems like the Lorenz or Rössler systems, to demonstrate the broader applicability of their method?

3. The paper mentions comparisons with RNNs, but it does not specify which types of RNNs were used in the experiments (in addition to LSTM). In recent years, various RNNs with specialized training algorithms have been developed to capture, model, and forecast time series data. Thus, to ensure a fair and comprehensive evaluation of NeRT's performance, the authors should include comparisons with these advanced RNN architectures, which are designed to capture **different types of temporal dependencies** in time series data. By benchmarking against **state-of-the-art RNNs** (e.g. [1,2,5]), the authors could provide clearer context for the advantages of their approach, thereby strengthening the validity of their claims regarding NeRT's superiority in handling periodic signals.
- Could the authors compare NeRT with some state-of-the-art RNNs, e.g., shPLRNN in [2], AL-RNNs in [5] or/and Sampled RNN in [1]?

---------------------------------------------------------------------
**References**

[1] Bolager, Erik Lien, Ana Cukarska, Iryna Burak, Zahra Monfared, and Felix Dietrich, "Gradient-free training of recurrent neural networks", arXiv preprint arXiv:2410.23467 (2024).

[2] Florian Hess, Zahra Monfared, Manuel Brenner, and Daniel Durstewitz, Generalized Teacher Forcing for Learning Chaotic Dynamics, In Proceedings of the 40th International Conference on Machine Learning. PMLR, (2023).

[3] Georgia Koppe, Hazem Toutounji, Peter Kirsch, Stefanie Lis, and Daniel Durstewitz, Identifying nonlinear dynamical systems via generative recurrent neural networks with applications to fmri, PLoS computational biology, 15(8):e1007263, (2019).

[4] Jonas Mikhaeil, Zahra Monfared, Daniel Durstewitz, On the difficulty of learning chaotic dynamics with RNNs, Advances in Neural Information Processing Systems, 35:11297-11312, (2022).

[5] Brenner, Manuel, Christoph Jürgen Hemmer, Zahra Monfared, Daniel Durstewitz, Almost-Linear RNNs Yield Highly Interpretable Symbolic Codes in Dynamical Systems Reconstruction, NeurIPS 2024, arXiv preprint arXiv:2410.14240 (2024).

**Questions:**

I) In Section 3.1, why was the activation function chosen as ReLU in Eq. (1)? Are there any guidelines or heuristics for selecting appropriate activation functions for different datasets or tasks?

II) Could the authors provide a more comprehensive and clearer comparison of their method with RNNs? Please see above ("Weaknesses").

**Details Of Ethics Concerns:**

No ethics concerns.

---

### Comment · Area_Chair_RqUX · 2024-11-23
**Request for Further Discussions**

Dear PC Members,

Thank you for your valuable comments during the review period, which raised many interesting and insightful questions. The authors have now posted their feedback, and I encourage you to review their responses and engage in further discussion if necessary.

I understand that you may have a busy schedule, but your additional input is highly appreciated, particularly for papers that have quite different opinions. Your contributions are crucial in ensuring a fair and well-rounded decision-making process.

Thank you once again for your continued support and dedication to ICLR.

Best regards,

---

### Meta-Review · Area_Chair_RqUX · 2024-12-19

**Metareview:**

This paper introduces a novel and effective framework for modeling periodic signals, offering significant improvements over existing methods. The reviewers acknowledged the method's originality and the advancements it brings. The paper is well-written, and the experiments, including the additional ones provided during the rebuttal, are thorough and sufficient.

While these strengths support the acceptance of the paper, the reviewers noted that the method's applicability is limited to periodic signals, which may reduce its broader appeal to the machine learning community. Consequently, there is no recommendation for a spotlight presentation.

**Additional Comments On Reviewer Discussion:**

From the very beginning, the reviewers recognized the novelty of this paper and acknowledged that it is well-written. The main concerns revolved around its limited application to periodic signals and the experimental evaluations. While the former is indeed a limitation, the latter was addressed effectively by the authors, who successfully convinced the reviewers with additional experiments. As a result, the reviewers increased their scores, leading to a final consensus.

---

### Decision · Program_Chairs · 2025-01-22

Accept (Poster)